# Belief Propagation Converges to Gaussian Distributions in Sparsely-Connected Factor Graphs

**Tom Yates** [1]  **Yuzhou Cheng** [1]  **Ignacio Alzugaray** [1]  **Danyal Akarca** [2]  **Pedro A.M. Mediano** [1]  **Andrew J. Davison** [1]

## Abstract

Belief Propagation (BP) is a powerful algorithm for distributed inference in probabilistic graphical models, however it quickly becomes infeasible for practical compute and memory budgets. Many efficient, non-parametric forms of BP have been developed, but the most popular is Gaussian Belief Propagation (GBP), a variant that assumes all distributions are locally Gaussian. GBP is widely used due to its efficiency and empirically strong performance in applications like computer vision or sensor networks – even when modelling non-Gaussian problems. In this paper, we seek to provide a theoretical guarantee for when Gaussian approximations are valid in highly non-Gaussian, sparsely-connected factor graphs performing BP (common in spatial AI). We leverage the Central Limit Theorem (CLT) to prove mathematically that variables' beliefs under BP converge to a Gaussian distribution in complex, loopy factor graphs obeying our 4 key assumptions. We then confirm experimentally that variable beliefs become increasingly Gaussian after just a few BP iterations in a stereo depth estimation task.

## 1. INTRODUCTION

Recent years have seen a surge in distributed and parallel computing architectures for spatial AI, from multi-core processors and GPUs to sensor networks and multirobot systems (Jouhari et al., 2023; Shorinwa et al., 2024). As problems become more distributed, there is a need for efficient algorithms that leverage local processing to solve global problems without centralized coordination, enabling scalability and resilience in complex environments.

BP is one such algorithm for distributed inference, consisting of local message passing where nodes interact only with their neighbours. First formalised as an algorithm for singly connected graphs by (Pearl, 1982), many more efficient variants of the algorithm have since emerged, such as Expectation Propagation (Minka, 2001) or Particle Belief Propagation (Ihler & McAllester, 2009). GBP (Weiss & Freeman, 1999a), a variant that assumes all distributions are Gaussian, dramatically improves tractability by efficiently representing beliefs with only their mean and variance.

BP's existing theory guarantees convergence to the exact posterior mean under strong assumptions, like Gaussian graphs (Weiss & Freeman, 1999a) or diagonally dominant covariances (Malioutov et al., 2006). While the validity of assuming all distributions are Gaussian in densely-connected graphs is well studied (Guo & Verdú, 2005; Donoho et al., 2009; Meng et al., 2015), the same cannot be said of the sparsely-connected graphs common to spatial AI. In dense graphs, individual factor nodes are assumed to have infinite degree, representing linear mixtures of all variables. The CLT thus occurs locally at a single factor node, as marginalising over the sum of incoming independent variable messages instantly produces a Gaussian outgoing message. In spatial AI, both variable and factors are finite, low-degree nodes, making this local, single-step CLT impossible. Despite this, practitioners routinely apply GBP for highly non-Gaussian robust estimation problems (Ortiz et al., 2021; Hug et al., 2024). These examples achieve strong empirical results with little theoretical justification. Our work intends to partially bridge this gap, offering guarantees common to spatial AI in which the Gaussian approximation is valid in highly non-Gaussian problems.

We offer a derivation showing that BP's variable beliefs converge to Gaussian distributions for non-Gaussian problems with highly uncertain data terms and complex graph topologies. Mathematical proofs and experimental results on a stereo depth estimation task demonstrate that the region of Gaussian convergence is empirically large, and so the Gaussian approximation can be applied to many probabilistic inference problems involving non-Gaussian elements. More specifically, our key contributions are:

---

[1]Imperial College London, London, United Kingdom [2]Cambridge University, Cambridge, United Kingdom. Correspondence to: Tom Yates <t.yates25@imperial.ac.uk>.

*Proceedings of the 43rd International Conference on Machine Learning*, Seoul, South Korea. PMLR 306, 2026. Copyright 2026 by the author(s).

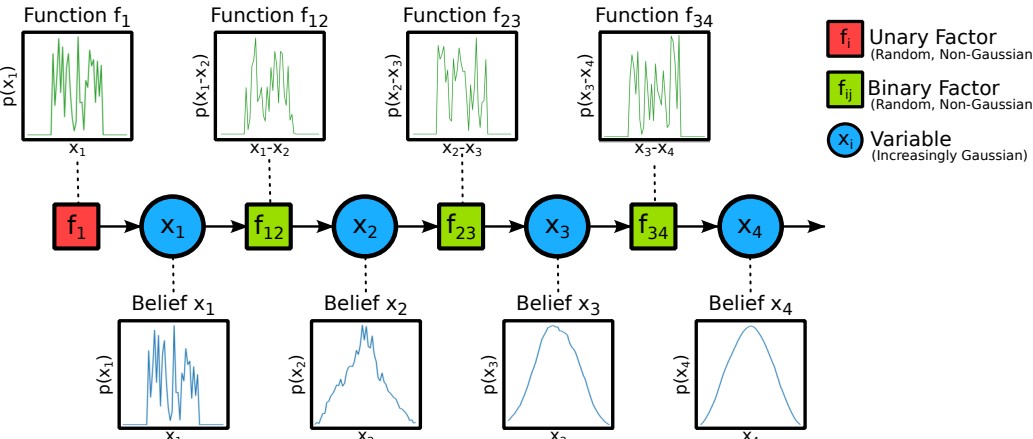

*Figure 1.* **Along a path of variables connected with random factors, beliefs become progressively Gaussian.** Variable $x_1$ has the only unary factor $f_1$ and inherits its distribution, but the variable beliefs of $x_2, x_3$ and $x_4$ are increasingly Gaussian, demonstrating the CLT's influence.

- **Unified Mechanism**: We reveal a mechanism by which Gaussian distributions naturally emerge in BP across sparsely-connected graph topologies.

- **Theoretical Derivation**: We provide the first derivation of BP's variable beliefs tending to Gaussian distributions in sparsely-connected graphs following Assumptions 1.1–1.4, detailed below.

- **Empirical Validation**: We validate these findings on synthetic and real stereo depth estimation tasks, demonstrating that the Gaussian assumption is valid for large regions of the graph even in non-Gaussian settings.

### 1.1. Scope & Assumptions

Our findings apply to sparsely-connected graphs with a fixed topology implementing synchronous BP. Such graphs are deterministic, so ergodic properties can be dismissed. For our proofs to hold, the graphs must obey:

**Assumption 1.1** (*Finite moments*). Functions for all factors in the graph have finite moments of all orders.

**Assumption 1.2** (*Lindeberg Condition*). The factor potentials along any path satisfy the Lindeberg condition (the variance contribution of any single factor asymptotically vanishes relative to the total path's variance).

**Assumption 1.3** (*Low-degree factors*). The graph contains only unary and binary factors.

**Assumption 1.4** (*Shift invariance*). Binary factors $f(x_i, x_j)$ only depend on their corresponding variables via their difference, $f(x_i, x_j) = g(x_i - x_j)$.

**Assumptions' Generality.** Assumptions 1.1 & 1.2 hold for almost all physical systems, where sensor limits ensure finite moments and redundancy prevents any single factor from dominating the total uncertainty. Intuitively, assumption 1.2 simply excludes physical scenarios such as a broken sensor with variance orders of magnitude larger than its neighbours, which would drown out the smoothing effect of convolution and prevent Gaussian convergence.

While assumptions 1.3 & 1.4 appear restrictive, they cover the vast majority of spatial AI problems. Regarding assumption 1.3, (Potetz & Lee, 2008) showed that linear high-degree factors (ternary or more) of the form $\phi(\mathbf{x}) = g(\mathbf{x} \cdot \mathbf{v})$ can be exactly decomposed to unary/binary factors. Although not tested in this work, our results are expected to extend directly to such high-degree factors.

Assumption 1.4 is stated for Euclidean vector spaces ($x_j - x_i$), but generalizes to Lie groups (e.g. $SO(3)$, $SE(3)$) via left-invariant potentials, which map to vector addition in the Lie Algebra (tangent space) (Solà et al., 2018). Consequently, although not explored here, a promising direction for future work is to explore where a tangent-space approximation in Lie Groups will break our results, and therefore when our results extend to the relative pose factors common in SLAM (see Appendix A).

Many spatial AI problems can be designed as graphs satisfying these assumptions, like image pixel estimation, robot odometry localisation, and networks of range-only sensors.

## 2. KEY INTUITIONS

Our work provides a theoretical validation of GBP's Gaussian assumption in uncertain regions of sparsely-connected, non-Gaussian graphs that follow assumptions 1.1–1.4. This encourages the use of GBP over more expensive

non-parametric methods like Particle-based BP (Ihler & McAllester, 2009) in such regions.

Figure 1 highlights a toy example of variable beliefs converging to a Gaussian distribution for a set of variables $x_1$ ... $x_4$ linked in a path-shaped factor graph. This is a simple but common set-up in engineering, that could for example represent tracking the position over time of a mobile robot with noisy odometry sensors.

Under a set of reasonable assumptions (detailed in Section 1.1), we prove that BP messages converge to a Gaussian distribution in long paths. This is because message passing at factors is equivalent to convolution, and by the CLT repeated convolution will drive distributions to a Gaussian in the asymptotic limit as path length from the closest data source (prior factor) tends to infinity.

Given tree and loopy graphs under our assumptions can be unwrapped as convolutional paths interspersed by multiplication, we extend this idea to trees and loopy graphs by showing that multiplication doesn't reduce Gaussianity in near-Gaussian distributions, and so near-Gaussian BP messages converge to a Gaussian distribution.

Empirically, Section 5 then validates this convergence mechanism across a range of topologies, exploring convergence rate, the effect of node degree and prior factor uncertainty. This is followed by results for a stereo depth estimation problem with highly non-Gaussian prior factors at every variable in a tight, loopy graph. We see Gaussian distributions emerge at the majority of variables apart from those with the strongest non-Gaussian prior factors, empirically showing this near-Gaussian constraint to be weak. Accordingly, we show variable beliefs' convergence to a Gaussian in a surprisingly broad range of settings, justifying GBP for many non-Gaussian problems.

# 3. PRELIMINARIES

## 3.1. Factor Graphs

A factor graph $G = (V, F)$ is a bipartite graph with variable nodes $V$ and factor nodes $F$ that represent the factorization of a joint probability distribution.

$$p(x_1, \ldots, x_n) \propto \prod_a f_a(x_a)$$

Variable nodes are associated with the marginal posterior distribution (belief) of their state. The factors are defined by their function potential $f$, capturing dependencies between connected variables. An example of a factor graph can be seen in Figure 1. Dellaert & Kaess (2017) provide a comprehensive overview of factor graphs and their applications.

For this work, all prior factors are unary (single neighbour) and are used to represent a data term. Binary factors (two

neighbours, pairwise) are considered *smoothing* if they constrain neighbouring variables to be close in value.

## 3.2. Belief Propagation

Belief Propagation is an entirely local algorithm for probabilistic inference on factor graphs, with a recent resurgence in prominence due to its suitability for highly distributed implementation. Inference proceeds via local message passing between nodes. Each message is a marginal probability distribution in the state space of a single variable. For a detailed description, see (Bishop, 2007) or (Ortiz et al., 2021).

We apply standard Sum-Product updates: variable-to-factor messages are the product of incoming messages, $m_{i \to a}(x_i) \propto \prod_{c \in \mathcal{N}(i) \backslash a} m_{c \to i}(x_i)$, and factor-to-variable messages are the marginalization of the factor potential, $m_{a \to i}(x_i) \propto \int f_a(X_a) \prod_{j \in \mathcal{N}(a) \backslash i} m_{j \to a}(x_j) dx_j$. A variable's belief is the product of its most recent incoming messages, $b(v) = \prod_{f \in n(v)} m_{f \to v}(v)$.

A graph is a singly connected tree if for any pair of nodes there exists at most one path between them. In trees, exact marginal inference at all nodes can be achieved with a single forward and backward pass of messages (from leaf nodes to root and back again). Note that while BP is exact on trees, we apply the same operations iteratively in Loopy BP on graphs with cycles.

## 3.3. Characteristic Functions & Cumulants

Given a discrete random variable $X$ with probability distribution $p(x)$, the n-th moment is defined as the expected value of $X^n$:

$$M_n = \mathbb{E}[X^n] \tag{1}$$

A probability distribution's moments can be used to define it, however to be exact it requires the moments' infinite series. A more compact representation is $X$'s characteristic function $\phi_x(t)$, which is defined as:

$$\phi_x(t) = \mathbb{E}[e^{itX}] \tag{2}$$

$\phi_x(t)$ is effectively the Fourier transform of the distribution, and uniquely defines it. If two random variables have the same $\phi(t)$, they must have the same probability distribution.

Moreover, we can also define a more useful property than moments, known as *cumulants*. The cumulant generating function is defined as:

$$K_x(t) = \ln(\phi_x(t)) \tag{3}$$

To better understand Equation (3), we can use the Taylor expansion of $\ln(1 + u) = u - \frac{u^2}{2!} + \ldots$, where $u = \phi_x(t) - 1$

to get an expression for $K_x(t)$ in terms of powers of $it$:

$$K_x(t) = (it\mathbb{E}[X] + \frac{(it)^2}{2!}\mathbb{E}[X^2] + ...)$$
$$- \frac{1}{2}(it\mathbb{E}[X] + \frac{(it)^2}{2!}\mathbb{E}[X^2] + ...)^2 + ...$$

If we expand the brackets and group by powers of $it$, we can define the cumulants $\kappa_n$ as the coefficients of $\frac{(it)^n}{n!}$ in the Taylor Expansion of $K_x(t)$:

$$K_x(t) = \frac{\kappa_1}{1!}(it)^1 + \frac{\kappa_2}{2!}(it)^2 + ...$$

By equating powers of $it$ in our expressions for $K_x(t)$, we see cumulants are a polynomial of low order moments:

$$\kappa_1(t) = \mathbb{E}[X] = \mu$$
$$\kappa_2(t) = \mathbb{E}[X^2] - (\mathbb{E}[X])^2 = \text{Var}(X)$$
$$\kappa_3(t) = \mathbb{E}[X^3] - 3\mathbb{E}[X]\mathbb{E}[X^2] + 2(\mathbb{E}[X])^3$$

(Hall, 1992) cover cumulants in detail. The characteristic function of a Gaussian distribution is:

$$\phi_G(t) = e^{it\mu - \frac{1}{2}\sigma^2 t^2} \tag{4}$$

so that for a Gaussian distribution, $\kappa_1 = \mu, \kappa_2 = \sigma^2$ and $\kappa_n = 0$ for all $n \geq 3$. Therefore, for any characteristic function $\phi(t)$, we can say $\kappa_{1,2}$ represent a Gaussian distribution with the same mean and variance, and the higher order cumulants $\kappa_{3,4,...}$ represent deviations from that Gaussian.

Finally, we must introduce the concept of *standardised cumulants*, defined as $\hat{\kappa}_n := \kappa_n / \sigma^n$ for $n \geq 3$, where $\sigma$ is the standard deviation of the distribution. Under convolution of independent variables, raw cumulants add but standardised cumulants decay at the rate of $m^{1-n/2}$ (demonstrated by . $m$ is the number of convolutions, and $n$ is the order of the cumulant, and this decay is what drives the distribution to a Gaussian. More formally, for i.i.d. random variables $X_1, \ldots, X_m$ with variance $\sigma^2$,

$$\hat{\kappa}_n\Big(\sum_{i=1}^{m} X_i\Big) = m^{1-\frac{n}{2}}\hat{\kappa}_n(X_1), \qquad n \geq 3. \tag{5}$$

## 4. MATHEMATICAL RESULTS

In this Section we present our main theoretical results, with proofs provided in the Appendices C–G.

### 4.1. Path Graphs

First, we consider a set of variables $X_1, ...X_n$ connected in a path by pairwise factors. The first variable $X_1$ has a single prior factor attached (Figure 2a). Paths are a special case of tree graphs, and so this graph can be solved with a single forward and backward pass (Section 3.2). Since the prior factor is the only source of data in the graph, we disregard the backward pass and consider only the forward pass.

In this topology, the belief at any subsequent variable $X_j$ is equal to the incoming forward message, so we analyse the evolution of this message to determine the beliefs' distributions. As the message propagates, uncertainty accumulates - each pairwise factor introduces additional noise defined by its potential function. Under Assumption 1.4 (shift invariance), BP's message update becomes a convolution of these potentials. Mathematically, since the convolution of Probability Density Function (PDF)s corresponds to the addition of independent random variables, we model the belief at $X_j$ as the random variable sum $X_j = X_1 + \sum_{k=1}^{j-1} Y_k$, where $Y_k$ represents the distribution of the $k$-th pairwise factor's potential. We get straight to our first building block:

**Theorem 4.1** (Path Convergence). *Under assumptions 1.1– 1.4, the belief of a variable in a pairwise path graph converges to a Gaussian distribution as its topological distance from the prior factor increases. (Proof in Appendix C)*

*Intuition:* The message update along the path is mathematically equivalent to a convolution of PDFs; by the CLT, this repeated convolution drives the message distribution toward a Gaussian.

**Justification of Independence via Convolution.** A critical requirement for the CLT is the independence of the random variables. While the state variables $X_1, \ldots, X_n$ in the path are strongly correlated, their Gaussian convergence arises from the accumulation of factor potentials (representing sensor noise), not the variables. Under Assumption 1.4, the BP message update from $X_i$ to $X_{i+1}$ is a convolution of the incoming message with an independent factor potential: $m_{i+1}(x) = (m_i * f_i)(x)$. The convolution of two PDFs is then equivalent to the distribution of the sum of two independent random variables. Therefore, the belief at depth $n$ is identical to the distribution of a sum $S_n = X_0 + Y_1 + \cdots + Y_n$, where each $Y_k$ is an independent random variable distributed according to the $k$-th factor potential $f_k$. In Spatial AI, these would commonly represent the cumulative addition of noise from a series of independent sensor measurements. Because the algorithm is convolving independent noise kernels, the requirements for the Lindeberg-Feller CLT are satisfied.

### 4.2. Multiple Priors

A variable with no prior factor is mathematically equivalent to a variable connected to a uniformly distributed prior factor. A uniform prior acts as an identity element during the belief update, preserving the cumulant decay from pairwise convolution. Consequently, Theorem 4.1 extrapolates to

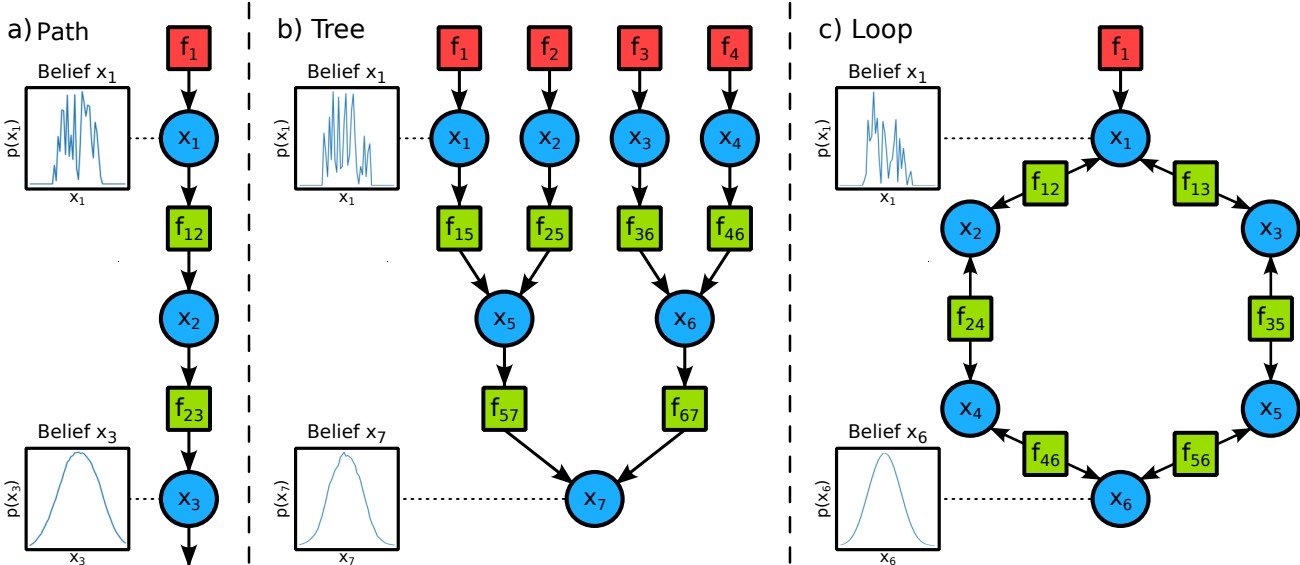

*Figure 2.* **BP drives variable beliefs to a Gaussian distribution for a range of graph topologies. a)** shows this for a singly connected path factor graph, connected by pairwise factors with one prior factor at the first variable. **b)** shows this for a singly connected tree graph where variables can be connected to more than two pairwise factors, with prior factors connected to the leaf nodes. In this case BP passes information from the leaf to root nodes. **c)** shows the same effect in a loopy graph connected by pairwise factors with a single prior factor attached to the first variable. BP is run iteratively here. In each of these different topologies, we see the variable beliefs converge to a Gaussian within a short distance of the closest prior factor.

graphs with prior factors at every variable, provided those priors have high uncertainty. We now formalize a theoretical baseline for a prior factor to act as a boundary condition that enforces its own non-Gaussian shape upon the local belief, preventing the CLT mechanism.

**Lemma 4.2.** *Let* $R = \sigma^2_{prior}/\sigma^2_{pairwise}$ *denote the ratio of local prior and pairwise factor variances. In a steady-state BP system on a path graph with priors at every variable, a variable's local prior will dominate the belief's shape & prevent the decay of non-Gaussian cumulants if:*

$$R \lesssim 6 \qquad (6)$$

*Intuition:* A confident prior ($R \lesssim 6$) acts as an anchor, forcing the belief to the prior's distribution regardless of the incoming messages' smoothing (proof in Appendix D).

Crucially, this bound reveals that the prior does not need to be more precise than the edge to dominate the belief shape. Because uncertainty accumulates across pairwise factors, the prior factor can have higher variance than a single edge and still dominate a path's variable belief.

This is derived under a worst-case, constructive interference of non-Gaussianity assumption. The empirical results of Section 5, where $R^*$ (the empirical $R$ threshold in graphs with priors at every variable) is slightly below 6, are therefore consistent with Appendix D which suggests destructive interference can allow Gaussian convergence at ratios lower than the idealised bound.

### 4.3. Tree Graphs

We now generalize to tree graphs – singly-connected factor graphs where variables may have more than two neighbours, or branching paths (Figure 2b). Unlike paths, where messages primarily undergo convolution, branching introduces a second operation: message multiplication at variable nodes (Section 3.2). Messages therefore undergo successive multiplication and convolution as they propagate from variable node to factor node. To prove Gaussian convergence, we must prove that this multiplication does not remove the Gaussianity accumulated by pairwise factor convolution. We can then generalise our findings to any tree graph topology.

To analyse this, we define $\varepsilon := \max_{n \geq 3} |\hat{\kappa}_n|$ as a quantitative measure of a distribution's "non-Gaussianity". We prove that, up to first order in $\varepsilon$, the multiplication of distributions at variable nodes does not increase their non-Gaussian component when $\varepsilon^2 \ll \varepsilon$:

**Lemma 4.3.** *Let* $p_a, p_b$ *have means* $\mu_a, \mu_b$, *variances* $\sigma^2_a, \sigma^2_b$, *and standardised cumulants* $\hat{\kappa}_{n,a}, \hat{\kappa}_{n,b}$ *with* $\max_{n \geq 3} |\hat{\kappa}_{n,\cdot}| \leq \varepsilon \ll 1$. *Let* $p_c(x) \propto p_a(x)p_b(x)$ *and define weights* $w_a := \sigma^2_a/(\sigma^2_a + \sigma^2_b)$, $w_b := \sigma^2_b/(\sigma^2_a + \sigma^2_b)$. *Then for* $n \geq 3$,

$$\hat{\kappa}_{n,c} = w_b^{n/2} \hat{\kappa}_{n,a} + w_a^{n/2} \hat{\kappa}_{n,b} + O(\varepsilon^2). \qquad (7)$$

*Intuition:* By linearizing the characteristic functions, we show that the non-Gaussian distortion introduced by multi-

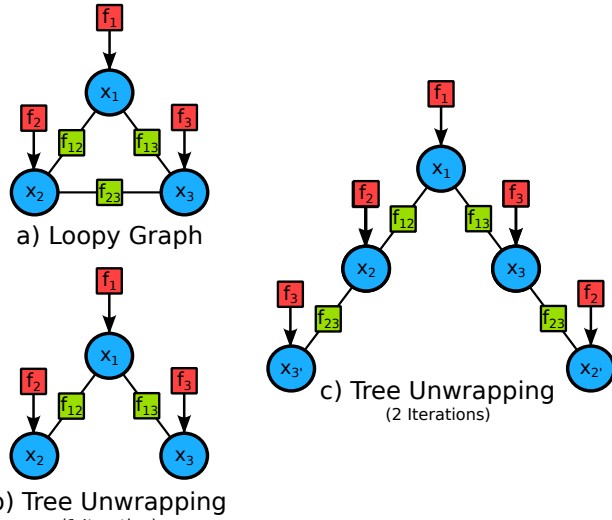

*Figure 3.* **Unwrapping a loopy factor graph into an equivalent tree graph. (a)** Loopy graph with three variables. **(b)** Depth-1 tree rooted at $x_1$: after one BP iteration, $b_{x_1}$ matches (a). **(c)** Extending the tree with identical local neighbourhoods yields matching beliefs for $x_1, x_2, x_3$ after one iteration; after two iterations, $x_1$ in (c) remains consistent with (a).

plication is of second order $\mathcal{O}(\varepsilon^2)$, making it asymptotically negligible compared to the first-order smoothing driven by convolution (proof in Appendix E). From this Lemma we can develop our next theorem:

**Theorem 4.4.** *Under Assumptions 1.1–1.4, the belief of a variable in a pairwise tree graph converges to a Gaussian distribution with topological distance from its closest prior factor, provided the message non-Gaussianity $\varepsilon$ satisfies $\varepsilon^2 \ll \varepsilon$. (Proof in Appendix F)*

### 4.4. Loopy Graphs

Finally, we consider the case of loopy graphs. We adopt (Weiss, 2000)'s computation trees to analyse Gaussian convergence and develop our last theorem.

**Theorem 4.5.** *Under Assumptions 1.1–1.4, variable beliefs in a loopy pairwise graph converge to a Gaussian distribution with topological distance from the closest prior factor, subject to the same $\varepsilon$ conditions as Theorem 4.4. (Proof in Appendix G).*

*Intuition:* Using the computation tree equivalence (Weiss, 2000), we show that messages in a loopy graph are identical to those in an unwrapped tree, reducing the problem to the convergence case established in Theorem 4.4.

Notably, loopy graphs may mean noise sources are traversed repeatedly and become correlated, violating the independence assumption of Theorem 4.1. However, synchronous BP unwraps the graph into a computation tree, which treats every reused factor as an independent random variable. This

incorrect independence assumption in loop BP's operator sequence explains why it is only an approximate algorithm, and why the message is driven to a Gaussian via the CLT.

Between Theorems 4.1 (paths), 4.4 (trees) & 4.5 (loopy graphs), we can then generalise our claim across graph topologies, such that for *any* graph following assumptions 1.1–1.4, the variable beliefs will converge to a Gaussian distribution with topological distance from the closest prior factor.

Moreover, Lemma 4.2 defines the 'exclusion zone' where the small-$\epsilon$ assumption fails for path graphs. Our convergence proofs (Theorems 4.4 & 4.5) apply to the paths between such anchors. Once a message leaves the higher-confidence regime ($R > 6$ in paths), the CLT dilution of Theorem 4.1 rapidly drives $\epsilon$ down, allowing the dynamics to enter Lemma 4.3's operating zone.

## 5. EXPERIMENTAL RESULTS

Our theoretical analysis proceeds in two phases. First, we verify the mechanism of convergence in synthetic experiments to analyse Gaussian convergence rates, specifically isolating the sensitivity to topology, prior strength and factor heterogeneity (Fig. 4a–c, Fig 5). Second, we deploy the proposed method on a distributed stereo depth estimation task. This real-world application serves to both stress-test our bounds under non-convex conditions (Fig. 4d) and demonstrate practical performance gains (Fig. 6). For clarity, our goal is not to demonstrate SOTA performance, but to evaluate the fidelity of Gaussian inference against non-parametric inference on identical factor graphs.

We use two metrics to quantify success. To measure Gaussian *emergence* - this paper's core claim - we calculate the Kullback-Leibler (KL) divergence ($D_{KL}$) between the belief of BP and its best-fit Gaussian approximation. Low KL divergence indicates the variable's belief has naturally evolved into a Gaussian shape. To measure the accuracy of our stereo depth estimate, we report the MSE of the solved disparities against the ground truth for BP and GBP. We average the results over random seeds and include standard deviation confidence intervals where applicable. Parameter values and implementation details are in Appendix I.

We first validate the theoretical mechanism driving Gaussian emergence. Figure 4(a) compares convergence rates with respect to distance across topologies. We see that all graph topologies converge to a Gaussian distribution (defined as $D_{KL} < 0.02$) rapidly, within 3 variable-variable hops. Moreover, we derive the $O(1/\sqrt{N})$ decay of higher-order cumulants in Appendix B via Berry (1941); Esseen (1945)'s work. This is consistent with Figure 4(a), where the KL divergence—which scales with the square of the cumulants (Hall, 1992)—approximately exhibits $O(1/N)$

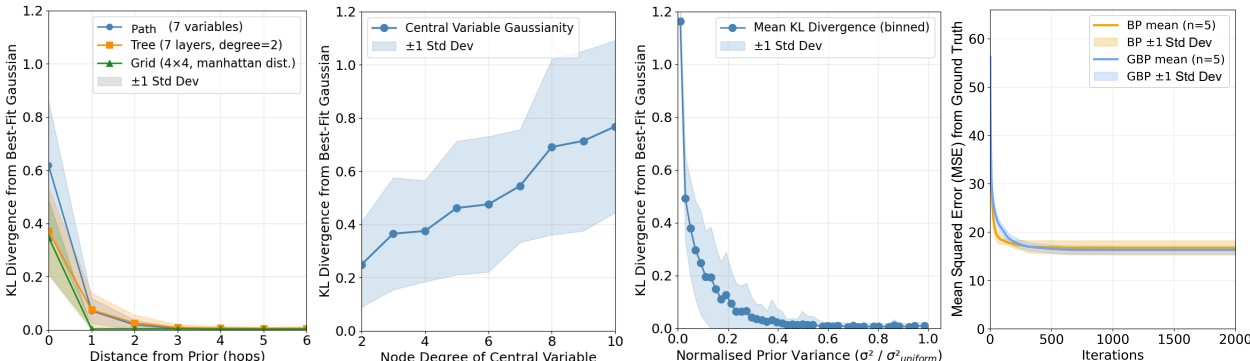

*Figure 4.* **Empirical Validation of the Convolutional CLT Mechanism.** (a) Convergence Rate: In sparse graphs, Gaussianity is driven by topological depth (convolution). All graphs converge to a Gaussian ($D_{KL} < 0.02$) within 3 hops, and Loopy grids (green) converge to a Gaussian faster than trees (orange) or paths (blue) due to multiple feedback paths. (a) Node Degree: In a Star graph without convolutional depth, increasing the node degree increases the non-Gaussianity of the central belief. This confirms that unlike in dense "large system" limits (Guo & Verdú, 2005), Gaussianity in sparse spatial AI graphs arises from path depth, not nodal averaging. (c) Prior Factor Uncertainty: Validating Lemma 4.2, variable beliefs remain non-Gaussian ($D_{KL} > 0.02$) only when anchored by high-confidence priors (Normalized Variance $< 0.5$). As prior uncertainty increases, the convolution mechanism dominates, driving $D_{KL}$ to zero. (d) GBP Accuracy: Despite the non-Gaussian nature of the stereo depth problem (Cones scene), GBP serves as an accurate surrogate optimiser for non-parametric BP, converging to the same final Mean Squared Error (MSE) with negligible approximation error.

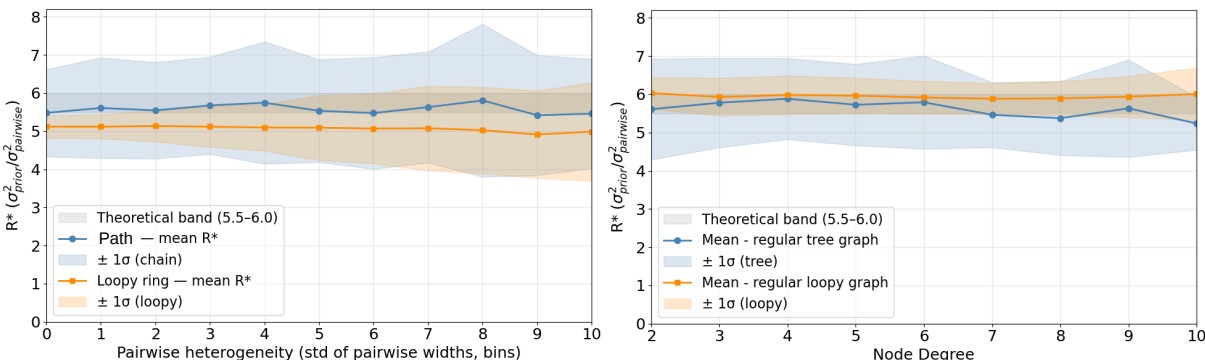

*Figure 5.* **Robustness of the Empirical Convergence Threshold ($R^*$) to Factor Heterogeneity & Graph Structure.** (Left) The empirical threshold $R^*$ remains close to Lemma 4.2's theoretical band of 6.0 across path and loopy topologies. This stability holds as the pairwise factor potentials become highly heterogeneous, confirming that the threshold relies on local variance ratios rather than graph-wide homogeneity. (Right) For both regular tree and loopy graphs, $R^*$ varies only weakly as node degree increases from 2 to 10.

decay.

Crucially, Figure 4(b) isolates the effect of node degree by analysing a Star graph with no convolutional depth. In this case all outer variables have highly non-Gaussian prior factors, and are each connected to a single central variable with no prior factor, who's belief is solely influenced by its neighbours. We observe that increasing the node degree ($N$) actually increases non-Gaussianity, confirming that Lemma 4.3 does not apply for multiplication of highly non-Gaussian distributions. This empirical result distinguishes our "sparse" regime from the "large system limit" in dense coding theory (Guo & Verdú, 2005), confirming that in spatial AI, Gaussianity is driven by path depth (convolution) and not nodal averaging.

We next investigate the conditions where the Gaussian approximation fails. Using a path graph with non-Gaussian prior factors at every variable, we vary the priors' uncertainty from a delta spike to uniform distribution. Figure 4(c) plots the final variable belief's KL divergence against the prior's variance. We see a sharp "Exclusion Zone" for high-confidence priors, where the non-Gaussian shape of the data anchors the belief, preventing the CLT mechanism. This phenomenon is visualized physically in the Cones scene (Figure 6). The heatmap reveals that high KL divergence (red) persists exclusively at high-contrast object edges where photometric evidence is strong. In contrast, textureless regions (green) lack strong priors, allowing the smoothing factors to dominate and drive the belief to a Gaussian, consistent with Lemma 4.2.

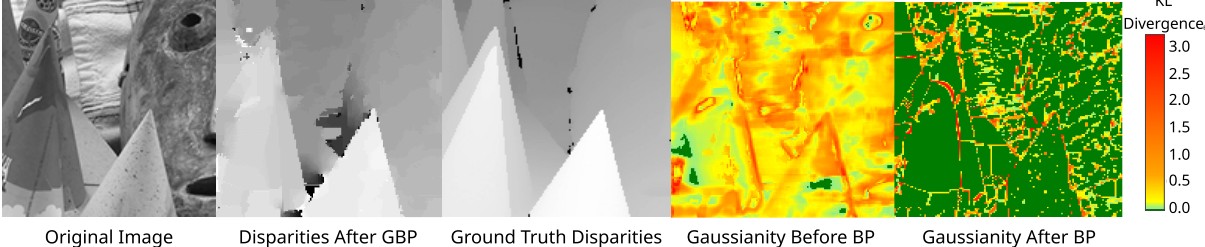

Original Image     Disparities After GBP     Ground Truth Disparities     Gaussianity Before BP     Gaussianity After BP

*Figure 6.* **Stereo Depth Estimation: BP tends to Gaussian beliefs under weak priors.** In the Cones scene (Scharstein & Szeliski, 2003), BP yields Gaussian-like variable beliefs ($D_{KL} < 0.02$, green) for pixels with high-variance priors, while confident-prior pixels such as edges remain non-Gaussian ($D_{KL} > 0.02$, red).

To assess $R$'s sensitivity to factor heterogeneity and topology, we defined an empirical $R^*$ threshold as the smallest value of $R$ for which the measured belief meets the same KL criterion used elsewhere ($D_{KL} < 0.02$). On heterogeneity, we measured $R^*$ using the centre of a path and ring of 10 variables with increasing heterogeneity in pairwise factors (Fig 5a). On topology, we find that $R^*$ varies only weakly with node degree in both topology families, with a modest uplift for loopy graphs (Fig 5b). Taken together, these results suggest that once messages are near-Gaussian after a few hops (Fig 4a), the $R$ threshold is reasonably robust.

We next evaluate Gaussian emergence on distributed stereo depth estimation (the 'Cones' scene), solving for pixel disparities $d$. This setup presents a challenging test case due to the tight loopy grid structure and highly non-Gaussian priors. Crucially, the formulation satisfies Assumptions 1.1–1.4, ensuring our theory is applicable. Figure 4(d) validates the approximation in this regime: despite the non-convex landscape, GBP closely tracks the non-parametric BP baseline with negligible MSE penalty. Qualitatively (Fig. 6), GBP recovers dense depth maps close to ground truth. While the Gaussian approximation induces minor smoothing at occlusion boundaries, it resolves global structure without the computational cost of particle-based methods.

In dense stereo, we recover physical depth $Z$ from two rectified images separated by a baseline $B$. We estimate the disparity $d$—the horizontal pixel shift between corresponding points $(u, v)$ and $(u - d, v)$—in the left and right images. Depth is then recovered geometrically via $Z = fBd^{-1}$, where $f$ is the camera's focal length.

We represent the image as a pairwise factor grid where variable $x_i$ denotes pixel $i$'s disparity. Each variable connects to a unary prior factor derived from local photometric matching costs, and also to its four neighbors via smoothing binary factors that encourage neighbouring variables to have similar beliefs (adjacent pixels are expected to have similar depths). To preserve depth discontinuities, we remove binary factors between pixels with high intensity differences,

preventing smoothing across image edges. This creates a loopy graph with priors at every variable node, serving as a difficult stress-test for our theory.

We evaluate the Middlebury 'Cones' scene (Scharstein & Szeliski, 2003), with additional scenes in Appendix H. We omit Variational Inference and Expectation Propagation baselines, as their fixed distributional assumptions (e.g. Gaussian) would mask the natural emergence of Gaussianity we seek to measure. Instead, we use Loopy BP as a non-parametric reference, allowing the posterior shape to evolve naturally. Comparing this against GBP isolates the information loss caused strictly by the Gaussian approximation.

The results of Figure 6 reveal a clear split in variables' belief shape that is governed by prior uncertainty. In textureless regions, where exact photometric matching is difficult and variables have weak priors (high uncertainty), we observe the predicted Gaussian convergence. The smoothing effect of convolution dominates, driving KL divergence to near-zero, validating Theorem **??**. Conversely, at image edges where photometric matching produces priors that are confident and highly non-Gaussian, the KL divergence remains high, as these non-Gaussian priors dominate the variable belief's shape over its incoming messages. Crucially, this local non-Gaussianity does not destabilise the global solution.

Moreover, the results empirically demonstrate how wide Lemma 4.3's near-Gaussian constraint is, showing that only the highest-confidence, non-Gaussian prior regions (e.g. edges or strong image features) remain non-Gaussian after optimisation. We also see that although we take the asymptotic limit as path length from strong prior factors tends to infinity in Section 4, empirically we see very low KL divergence values in remarkably short path lengths from strong image features like edges or colour disparities.

We see in Appendix H that Gaussian emergence is consistent across all scenes tested. Gaussian heat maps show convergence to Gaussian distributions in textureless regions, while preserving the non-Gaussian distributions of highly textured regions (Fig 6, Appendix Fig 9). Moreover, the

total Mean Squared Error converges to remarkably close values between BP and GBP across all scenes. These results imply that variable beliefs are dominated by Gaussian interactions in BP for a range of practical Spatial AI tasks.

On parameter sensitivity, we found Gaussian emergence to be robust to variations in graph size and factor shape, provided a few key conditions are met. First, beliefs' discretisation resolution must be sufficient to faithfully represent a Gaussian (e.g. $\geq 20$ bins). Second, the graph must have enough topological depth for the required message passing to enable the CLT (e.g. 2-3 hops, Fig 4a). Third, although the asymptotic limit of the CLT allows variance to accumulate indefinitely, real discrete distributions have finite support; therefore, pairwise potentials must be kept narrow to avoid distribution truncation.

Beyond these practical requirements, the main sensitivity is the relative strength of the local prior and pairwise factors, as captured by $R$. Our new topology experiment suggests that once messages have entered the near-Gaussian regime, the empirical threshold depends only weakly on topology.

# 6. RELATED WORK

BP was originally formalised by Pearl (1982) as an exact inference algorithm for singly-connected (tree) graphs. While the algorithm is only exact for trees, Frey & MacKay (1997) showed that "Loopy" BP could achieve near-Shannon-limit performance on sparsely-connected graphs, specifically Low-Density Parity Check codes. This empirical success encouraged the use of BP in graphs with cycles despite the lack of theoretical guarantees. Weiss & Freeman (1999b) later validated BP's numerical accuracy in Gaussian graphs by proving that loopy BP calculates exact posterior means, paving the way for the more efficient GBP variant.

Although there have been significant theoretical advancements in Loopy BP's convergence conditions (Ihler et al., 2005; Koehler, 2019), existing guarantees for general graph topologies still require strong assumptions, such as Gaussian graphs (Weiss & Freeman, 1999b), or diagonally dominant covariances (Malioutov et al., 2006).

If we restrict the topology to be densely-connected, BP's theoretical properties are well studied. In the domain of communications and coding theory, researchers such as Guo & Verdú (2005) and Donoho et al. (2009) successfully proved Gaussian convergence for BP in non-Gaussian settings. However these results rely on the "large system limit" of densely-connected graphs, where the CLT applies via the linear summation of a massive number of neighbours ($Degree \rightarrow \infty$) at the factor nodes (representing signal interference). Their key assumption is for all variables to be connected to all factors, taking the limit as the graph size extends to $\infty$ such that the cumulative interference at

each node becomes Gaussian. These arguments fail for the sparsely-connected, low-degree graphs common in spatial AI, where variable beliefs are driven by local products of constraints rather than massive averaging.

Despite this, spatial AI practitioners regularly use GBP in highly non-Gaussian problems, such as robust estimation (Ortiz et al., 2021; Hug et al., 2024) or distributed robotics (Jiang, 2024), achieving strong empirical results without theoretical justification. We bridge this gap by proving that Gaussianity emerges in these sparse, low-degree settings through a novel mechanism: repeated convolution along deep paths ($depth \rightarrow \infty$).

# 7. CONCLUSION

We have provided a derivation identifying the conditions in which GBP's Gaussian assumption is valid for highly non-Gaussian problems in sparsely-connected graphs common to spatial AI. Our analysis, based on cumulant expansions and CLT arguments, shows that repeated convolution and multiplication of near-Gaussian distributions drive higher-order cumulants to decay, leaving the Gaussian term dominant in the distribution. These results suggest that GBP can be applied to a broader range of problems in graphical models and probabilistic reasoning than initially expected, and potentially justifies the use of hybrid inference between Gaussian and non-parametric methods.

There are several limitations to our work - in particular, our guarantees do not directly extend to:

1. Non-Euclidean state spaces (e.g. Lie Groups), except where tangent-space approximations remain valid

2. Factors of degree greater than unary/binary, beyond linear cases of the form $\phi(\mathbf{x}) = g(\mathbf{x} \cdot \mathbf{v})$ (which reduce to recursive 1D convolutions via auxiliary variables)

3. Asynchronous or adaptive message schedules

4. Loopy graphs that do not converge to steady state beliefs under synchronous BP

5. Graphs where factor distributions are not known, so $R$ cannot be evaluated directly

Accordingly, our work opens several useful goals for further research, such as analysis of the speed and stability of cumulant decay under asynchronous BP schedules, or for graphs where not all pairwise constraints obey assumption 1.4 (shift invariance). Moreover, extending this cumulant analysis to uncertainties on Lie Groups presents a critical direction for future work in robust robot perception.

## Impact Statement

This paper presents work whose goal is to advance the field of Machine Learning. There are many potential societal consequences of our work, none which we feel must be specifically highlighted here.

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

# A. Extension of Assumption 1.4 to Lie Groups

While Assumption 1.4 (Shift Invariance) is stated for Euclidean vector spaces where the error is defined as $\mathbf{x}_j - \mathbf{x}_i$, this formulation is expected to generalise naturally to the Lie groups common in spatial AI, such as $SO(3)$ or $SE(3)$.

In the context of Lie groups, the shift-invariant assumption corresponds to *left-invariant potentials*. For two group elements $X_i, X_j \in \mathcal{G}$, the factor potential depends only on the relative group action:

$$f(X_i, X_j) = g(X_i^{-1} \circ X_j) \tag{8}$$

For distributions that are well-localised on the manifold (i.e., the distribution noise is small relative to the manifold curvature), the group operations can be mapped to the Lie Algebra $\mathfrak{g}$ (the tangent space at the identity) via the logarithmic map:

$$\boldsymbol{\xi}_{ij} = \mathrm{Log}(X_i^{-1} \circ X_j) \in \mathbb{R}^d \tag{9}$$

where $d$ is the degrees of freedom of the group. As detailed in Solà et al. (2018), operations on the manifold for these well-localised distributions are effectively equivalent to vector addition in the tangent space. Consequently, the random variable addition $S_n = X_0 + \sum Y_k$ described in Section 4.1 maps to the summation of tangent space vectors $\boldsymbol{\xi}_\Sigma \approx \sum \boldsymbol{\xi}_k$.

Therefore, the convolutional CLT mechanism derived in Theorem 4.1 applies directly to the tangent-space coordinates of these relative pose factors. This extends the validity of our Gaussian convergence results to the relative pose graphs standard in SLAM and robotics, but requires further work to understand how the tangent-space approximation affects convergence to a Gaussian.

## B. Remark on Convergence Rates.

The rate of Gaussian convergence is governed by the competition between the convolution and multiplication operations of the message updates.

During convolution at pairwise factors, the dominant (slowest-decaying) non-Gaussian cumulant, skewness, decays at a rate of $\kappa_3 \propto \mathcal{O}(d^{-1/2})$ by the Berry-Esseen theorem (Berry, 1941; Esseen, 1945). Here $d$ represents the path length to the closest prior factor. This establishes the baseline decay rate for non-Gaussianity: $\varepsilon \propto \mathcal{O}(d^{-1/2})$.

At variable nodes, this decay competes with the error term introduced by message multiplication. For a variable of degree $K$, Lemma 4.3 indicates that multiplying the incoming messages introduces a non-Gaussian perturbation bounded by the sum of second-order interactions: $\sum_{i=1}^{K} \mathcal{O}(\varepsilon_i)^2 \leq K \cdot \mathcal{O}(\varepsilon_{max}^2)$, where $\varepsilon_{max}$ is the largest standardized higher-order cumulant in the distributions being multiplied.

Substituting the convolution decay rate into this expression reveals that the added non-Gaussian perturbation scales as $\mathcal{O}(K/d)$. Because this corrupting perturbation $\mathcal{O}(d^{-1})$ decays quadratically faster than the remaining non-Gaussianity $\mathcal{O}(d^{-1/2})$, the Gaussian convergence mechanism is asymptotically stable as $d \to \infty$.

Intuitively, this implies that near strong priors or high-degree nodes where the ratio $K/d$ is significant, variable beliefs will be non-Gaussian. However, as path length increases, the smoothing effect of convolution strictly dominates the error from multiplication. In spatial AI, graphs are typically sparse (low constant $K$). Consequently, the regime where convolution dominates ($d \gg K$) is reached at short topological distances, ensuring the non-Gaussian boundary layer remains thin.

## C. Proof of Theorem 4.1

In this Section, we provide the complete derivation for Theorem 4.1. We analyse the propagation of information along a singly-connected path of variables to demonstrate how the repeated convolution of factor potentials acts as a smoothing mechanism. By establishing a mathematical equivalence between this message passing process and the summation of independent random variables, we leverage the CLT to prove that the belief distribution asymptotically converges to a Gaussian shape as the path length increases.

Consider the message $m_{i \to i+1}(x_{i+1})$ sent from variable $x_i$ to $x_{i+1}$ along the path. By the Belief Propagation update rule, the message is the product of the incoming message and the local factor, integrated over the source variable:

$$m_{i \to i+1}(x_{i+1}) \propto \int f_{i,i+1}(x_i, x_{i+1}) \, m_{i-1 \to i}(x_i) \, dx_i \tag{10}$$

Under Assumption 1.4 (Shift Invariance), the factor potential depends only on the difference between variables: $f_{i,i+1}(x_i, x_{i+1}) = g_i(x_{i+1} - x_i)$. Substituting this into the update equation yields:

$$m_{i \to i+1}(x_{i+1}) \propto \int g_i(x_{i+1} - x_i) \, m_{i-1 \to i}(x_i) \, dx_i \tag{11}$$

This is the precise definition of a convolution operation $m_{i \to i+1} = g_i * m_{i-1 \to i}$.

Since the convolution of PDFs corresponds to the addition of independent random variables, the belief (message) at a topological distance $n$ is proportional to the PDF of the sum:

$$S_n = X_1 + \sum_{k=1}^{n} Y_k \tag{12}$$

where $X_1$ is the random variable associated with the prior factor at the start of the path, and $\{Y_k\}$ are independent random variables distributed according to the potentials $g_k$.

We analyse the sum $\sum_{k=1}^{n} Y_k$. Let $\sigma_k^2$ be the variance of the $k$-th factor potential $Y_k$.

- By Assumption 1.1 (Finite Moments), $\sigma_k^2 < \infty$ for all $k$.

- By Assumption 1.2 (Lindeberg Condition), the variance of the partial sum $s_n^2 = \sum_{k=1}^{n} \sigma_k^2$ diverges ($s_n \to \infty$) such that no single factor dominates the total variance.

Thus, the conditions for the Lindeberg-Feller CLT (Durrett, 1996) are satisfied. The standardized sum converges in distribution to a standard normal:

$$\frac{\sum_{k=1}^{n}(Y_k - \mu_k)}{s_n} \xrightarrow{d} \mathcal{N}(0,1) \tag{13}$$

The total sum is $S_n = X_1 + \sum Y_k$. As the path length $n \to \infty$, the accumulated variance of the factors $s_n^2$ grows indefinitely. The contribution of the initial prior $X_1$ (which has fixed, finite variance) becomes negligible relative to the accumulated variance of the path.

By Slutsky's theorem, adding the independent constant-variance term $X_1$ does not alter the asymptotic distribution of the standardized sum. Consequently, the standardized cumulants of order $n \geq 3$ converge to 0, and the belief distribution converges to a Gaussian as $n \to \infty$.

# D. Proof of Lemma 4.2

In this Section, we provide the complete derivation for Lemma 4.2. We determine the conditions under which a local prior factor acts as a Non-Gaussian anchor in a path graph with priors connected to every node, effectively resetting the belief distribution and preventing the CLT from driving the variable to a Gaussian.

## 1. Steady-State Variance in a Homogeneous Path

Consider a variable $X_i$ in a steady-state path where every variable is connected with pairwise factors of variance $\sigma_e^2$ and every variable connects to a prior with variance $\sigma_p^2$. Let $\sigma_{ss}^2$ denote the steady-state marginal variance of the variable.

The steady-state precision is the sum of the incoming message precision (after pairwise convolution) and the local prior precision:

$$\frac{1}{\sigma_{ss}^2} = \underbrace{\frac{1}{\sigma_{ss}^2 + \sigma_e^2}}_{\text{Incoming Message}} + \underbrace{\frac{1}{\sigma_p^2}}_{\text{Local Prior}} \tag{14}$$

Rearranging to solve for the prior variance $\sigma_p^2$:

$$\frac{1}{\sigma_p^2} = \frac{1}{\sigma_{ss}^2} - \frac{1}{\sigma_{ss}^2 + \sigma_e^2} = \frac{\sigma_e^2}{\sigma_{ss}^2(\sigma_{ss}^2 + \sigma_e^2)} \tag{15}$$

Inverting this yields the relationship between the physical variances:

$$\sigma_p^2 = \frac{\sigma_{ss}^2(\sigma_{ss}^2 + \sigma_e^2)}{\sigma_e^2} \tag{16}$$

**Definition of Variance Retention ($\lambda$).** We define $\lambda$ as the fraction of variance "retained" or "remembered" after a message traverses an edge. This acts as the memory coefficient for the system:

$$\lambda := \frac{\sigma_{ss}^2}{\sigma_{ss}^2 + \sigma_e^2} \quad \in (0, 1) \tag{17}$$

**Definition of Noise Ratio ($z$).** It is convenient to parameterize the system by the ratio of the edge noise to the steady-state variance:

$$z := \frac{\sigma_e}{\sigma_{ss}} \tag{18}$$

Substituting this into Eq. (17), we find the identity relating $\lambda$ and $z$:

$$\lambda = \frac{1}{1 + z^2} \quad \implies \quad z = \sqrt{\frac{1 - \lambda}{\lambda}} \tag{19}$$

## 2. Evolution of Standardized Cumulants

We now track the evolution of the standardized cumulants $\hat{\kappa}_n = \kappa_n / \sigma^n$ through one update cycle. The cycle consists of two steps:

1. **Convolution (Factors):** The message gains edge noise. Raw cumulants add; standardized cumulants are diluted by the increased variance.

2. **Multiplication (Variables):** The message is multiplied by the prior. Standardized cumulants are weighted by their precision shares (Lemma 4.3).

Using the exact moment propagation derived in Lemma 4.3, the steady-state standardized cumulant $\hat{\kappa}_{ss}$ satisfies:

$$\hat{\kappa}_{ss} = \underbrace{\lambda^n \hat{\kappa}_{ss}}_{\text{Memory}} + \underbrace{\lambda^n \left(\frac{\sigma_e}{\sigma_{ss}}\right)^n \hat{\kappa}_e}_{\text{Pairwise non-Gaussianity}} + \underbrace{(1 - \lambda)^{n/2} \hat{\kappa}_p}_{\text{Prior non-Gaussianity}} \tag{20}$$

Here, the first term represents the decay of the previous state's shape (Memory) given $\lambda < 1$, while the latter two terms represent the non-Gaussianity introduced by the pairwise factor and the prior (non-Gaussianity).

## 3. The Strong Prior Condition

We define a **Strong Non-Gaussian Prior** as a regime where the standardised cumulant's steady state is dominated by the non-Gaussianity introduced by the prior and pairwise factors.

$$\text{Prior + Pairwise non-Gaussianity} > \text{Memory} \tag{21}$$

Substituting the coefficients from the steady-state equation:

$$(1 - \lambda)^{n/2} + \lambda^n \left( \frac{\sigma_e}{\sigma_{ss}} \right)^n > \lambda^n \tag{22}$$

We analyze this condition for the skewness ($n = 3$), as it is the slowest-decaying non-Gaussian component and thus sets the critical bound. Substituting $z = \sigma_e/\sigma_{ss}$:

$$(1 - \lambda)^{1.5} + \lambda^3 z^3 > \lambda^3 \tag{23}$$

## 4. Solving for the Bound

To solve this inequality, we express all terms as functions of $z$. Recall $\lambda = (1 + z^2)^{-1}$ and $(1 - \lambda) = z^2(1 + z^2)^{-1}$. Substituting these into the inequality:

$$\left( \frac{z^2}{1 + z^2} \right)^{1.5} + \left( \frac{1}{1 + z^2} \right)^3 z^3 > \left( \frac{1}{1 + z^2} \right)^3 \tag{24}$$

Multiply the entire inequality by $(1 + z^2)^3$ to clear the denominators:

$$(z^2)^{1.5}(1 + z^2)^{1.5} + z^3 > 1 \tag{25}$$

$$z^3(1 + z^2)^{1.5} + z^3 > 1 \tag{26}$$

Divide by $z^3$:

$$(1 + z^2)^{1.5} + 1 > z^{-3} \tag{27}$$

We define $u = z^2$ and look for the critical value $u^*$ that satisfies the equality:

$$1 + (1 + u)^{1.5} = u^{-1.5} \tag{28}$$

This equation can be solved numerically.

- At $u = 0.5$: LHS $= 1 + (1.5)^{1.5} \approx 2.837$. RHS $= (0.5)^{-1.5} \approx 2.828$.
- Since LHS $>$ RHS, the condition holds for $u \geq 0.5$.

Thus, the critical threshold is $z^2 \approx 0.5$.

**Mapping to Prior Weakness Ratio** ($R$). Finally, we relate this result back to the physical ratio $R = \sigma_p^2/\sigma_e^2$. Using Eq. (16), we divide by $\sigma_e^2$ to get:

$$R = \frac{\sigma_{ss}^2}{\sigma_e^2} \left( \frac{\sigma_{ss}^2}{\sigma_e^2} + 1 \right) \tag{29}$$

Substituting $\frac{\sigma_{ss}^2}{\sigma_e^2} = \frac{1}{z^2} = \frac{1}{u}$:

$$R = \frac{1}{u} \left( \frac{1}{u} + 1 \right) \tag{30}$$

Inserting the critical value $u \approx 0.5$:

$$R \approx \frac{1}{0.5} \left( \frac{1}{0.5} + 1 \right) = 2(2 + 1) = 6 \tag{31}$$

**Note on Conservatism:** This derivation assumes a worst-case scenario where the non-Gaussian cumulants from the prior and the pairwise factor align constructively (same sign). In practice, alternating signs may lead to destructive interference, allowing Gaussian convergence at ratios lower than $R = 6$. Therefore, $R > 6$ represents the theoretical stability boundary for the idealised path. For more complex graphs, topological parameters such as node degree (Fig 4b) may impose stricter requirements.

# E. Proof of Lemma 4.3

In this Section, we derive Lemma 4.3, which addresses the stability of Gaussian convergence at variable nodes where incoming messages are multiplied. Unlike convolution, which actively suppresses higher-order cumulants, the multiplication of distributions can theoretically amplify non-Gaussian features. By expanding the characteristic functions of the incoming messages, we prove that provided the distributions are near-Gaussian (where non-Gaussianity $\varepsilon$ is small), the perturbations introduced by multiplication are bounded by $\mathcal{O}(\varepsilon^2)$. This ensures that, in this regime, the smoothing effect of convolution strictly dominates any errors introduced.

The multiplication of PDF $p_c(x) \propto p_a(x)p_b(x)$ corresponds to the convolution of their characteristic functions $\phi_a(t), \phi_b(t)$ in the Fourier domain:

$$\phi_c(t) \propto (\phi_a * \phi_b)(t) = \int_{-\infty}^{\infty} \phi_a(\tau)\phi_b(t-\tau)d\tau. \tag{32}$$

The characteristic functions $\phi_a(t), \phi_b(t)$ can be defined in terms of their cumulants. For $j \in \{a,b\}$:

$$\phi_j(t) = e^{\sum_{n=1}^{\infty} \frac{\kappa_{n,j}}{n!}(it)^n} \tag{33}$$

From this expression, we can factor out a Gaussian core $\phi_{G,j}$

$$\phi_j(t) = (e^{\frac{\kappa_{1,j}}{1!}(it)^1 + \frac{\kappa_{2,j}}{2!}(it)^2})(e^{\frac{\kappa_{3,j}}{3!}(it)^3 + \frac{\kappa_{4,j}}{4!}(it)^4 + \cdots}) = \phi_{G,j}(t)e^{\sum_{n=3}^{\infty} \frac{\kappa_{n,j}}{n!}(it)^n} \tag{34}$$

And expand the higher-order cumulants term that represents non-Gaussian perturbations. For $j \in \{a,b\}$:

$$\phi_j(t) = \exp\left(i\mu_j t - \frac{1}{2}\sigma_j^2 t^2\right)\left[1 + \sum_{n\geq 3} \frac{(it)^n}{n!}\kappa_{n,j} + \mathcal{O}(\epsilon^2)\right] = \phi_{G,j}(t) + R_j(t). \tag{35}$$

Where $R(t) = \phi_{G,j}(t)\sum_{n\geq 3} \frac{(it)^n}{n!}\kappa_{n,j}$ is a remainder term capturing the higher-order cumulants. Substituting this into the convolution integral, we arrive at:

$$\phi_c(t) \propto \int_{-\infty}^{\infty} (\phi_{G,a}(\tau) + R_a(\tau))(\phi_{G,b}(t-\tau) + R_b(t-\tau))d\tau \tag{36}$$

$$= \int_{-\infty}^{\infty} \phi_{G,a}\phi_{G,b}d\tau + \int_{-\infty}^{\infty} \phi_{G,a}R_b d\tau + \int_{-\infty}^{\infty} R_a\phi_{G,b}d\tau + \int_{-\infty}^{\infty} R_a R_b d\tau \tag{37}$$

$$= I_1(t) + I_2(t) + I_3(t) + I_4(t) \tag{38}$$

(Note: we omit the function arguments $\tau$ and $t-\tau$ for brevity)

We focus on the terms in order. $I_1(t)$ represents the product of two Gaussian distributions, analysed by their convolution in the Fourier domain:

$$I_1(t) = \int \exp\left(i\mu_a\tau - \frac{1}{2}\sigma_a^2\tau^2\right)\exp\left(i\mu_b(t-\tau) - \frac{1}{2}\sigma_b^2(t-\tau)^2\right)d\tau. \tag{39}$$

This is a known, standard result - the product of two Gaussians gives a scaled Gaussian with mean $\mu_{new} = \frac{\mu_a\sigma_b^2 + \mu_b\sigma_a^2}{\sigma_a^2 + \sigma_b^2}$ and variance $\sigma_{new}^2 = \frac{\sigma_a^2\sigma_b^2}{\sigma_a^2 + \sigma_b^2}$. Evaluating the integral $I_1(t)$ therefore yields a Gaussian core with the resulting characteristic function:

$$\phi_{G,c}(t) \propto \exp\left(i\left(\frac{\mu_a\sigma_b^2 + \mu_b\sigma_a^2}{\sigma_a^2 + \sigma_b^2}\right)t - \frac{1}{2}\left(\frac{\sigma_a^2\sigma_b^2}{\sigma_a^2 + \sigma_b^2}\right)t^2\right) \tag{40}$$

We next focus on the first cross-term, $I_2(t)$, which convolves the Gaussian core of $a$ with the non-Gaussian perturbation of $b$:

$$I_2(t) = \int_{-\infty}^{\infty} \phi_{G,a}(\tau)\phi_{G,b}(t-\tau)\left[\sum_{n\geq 3} \frac{\kappa_{n,b}}{n!}(i(t-\tau))^n\right]d\tau. \tag{41}$$

The product of the two Gaussian cores, $\phi_{G,a}(\tau)\phi_{G,b}(t-\tau)$, can be factored exactly into a Gaussian core $\phi_{G,c}(t)$ with respect to $t$ and a conditional Gaussian density with respect to $\tau$:

$$\phi_{G,a}(\tau)\phi_{G,b}(t-\tau) = \phi_{G,c}(t) \cdot \mathcal{N}(\tau \mid \mu_\tau, \sigma_\tau^2). \tag{42}$$

The parameters of this conditional density are determined by the product of Gaussians formula:

$$\mu_\tau = \frac{\sigma_b^2}{\sigma_a^2 + \sigma_b^2}t = w_a t, \quad \sigma_\tau^2 = \frac{\sigma_a^2 \sigma_b^2}{\sigma_a^2 + \sigma_b^2}. \tag{43}$$

Substituting this into the expression for $I_2(t)$, we see $\phi_{G,c}(t)$ depends only on t and so can pull it outside the integral. The remaining integral then represents the expected value of the non-Gaussian perturbation under the conditional distribution $\mathcal{N}(\tau \mid \mu_\tau, \sigma_\tau^2)$:

$$I_2(t) = \phi_{G,c}(t) \sum_{n \geq 3} \frac{\kappa_{n,b}}{n!} i^n \mathbb{E}_\tau \left[ (t-\tau)^n \right]. \tag{44}$$

Let $Y = t - \tau$. Since $\tau \sim \mathcal{N}(w_a t, \sigma_\tau^2)$, the variable $Y$ is also Gaussian distributed with mean $\mu_Y = t - w_a t = (1 - w_a)t = w_b t$ and variance $\sigma_\tau^2$. The raw moments of a Gaussian variable $Y \sim \mathcal{N}(\mu_Y, \sigma_\tau^2)$ are given by a polynomial in $\mu_Y$:

$$\mathbb{E}[Y^n] = \mu_Y^n + \binom{n}{2}\mu_Y^{n-2}\sigma_\tau^2 + \cdots = (w_b t)^n + P_{n-2}(t), \tag{45}$$

where $P_{n-2}(t)$ contains terms of order $t^{n-2}, t^{n-4}, \ldots$ resulting from the variance. Substituting this expectation back into $I_2(t)$:

$$I_2(t) = \phi_{G,c}(t) \sum_{n \geq 3} \frac{\kappa_{n,b}}{n!} i^n \left[ (w_b t)^n + O(t^{n-2}) \right] \tag{46}$$

We define the "leading order" contribution as the term that preserves the power of $t$ (and thus maps $\kappa_{n,b}$ to $\kappa_{n,c}$). The variance corrections ($O(t^{n-2})$) contribute to cumulants of order strictly less than $n$.

By symmetry, the term $I_3(t)$ convolves the perturbation of $a$ with the core of $b$. The relevant mean is $w_a t$, yielding:

$$I_3(t) = \phi_{G,c}(t) \sum_{n \geq 3} \frac{\kappa_{n,a}}{n!} i^n \left[ (w_a t)^n + O(t^{n-2}) \right]. \tag{47}$$

The final term, $I_4(t)$, involves the convolution of two remainder terms ($R_a * R_b$). Since $R_a$ and $R_b$ are both $O(\epsilon)$, their convolution is $O(\epsilon^2)$. Combining $I_1$ (the core), $I_2$, $I_3$ and $I_4$ into the expansion for $\phi_c(t)$:

$$\phi_c(t) = \phi_{G,c}(t) \left[ 1 + \sum_{n \geq 3} \frac{(it)^n}{n!} (w_a^n \kappa_{n,a} + w_b^n \kappa_{n,b}) + \cdots + O(\epsilon^2) \right]. \tag{48}$$

The omitted terms "$\ldots$" include the variance corrections from $I_2(t)$ and $I_3(t)$ which affect lower-order $t$ terms. To identify the $n$-th cumulant $\kappa_{n,c}$, we match the coefficient of $\frac{(it)^n}{n!}$ in the log-expansion. At first order in $\epsilon$, this coefficient is uniquely determined by the leading terms derived above:

$$\kappa_{n,c} = w_a^n \kappa_{n,a} + w_b^n \kappa_{n,b} + O(\epsilon^2). \tag{49}$$

(Note: Technically, terms from $k > n$ in the sum could contribute to the $t^n$ coefficient via variance corrections. However, these are "mixing" terms that are negligible in the first-order separation of modes, or absorbed into the higher-order error term.)

We convert to standardized cumulants $\hat{\kappa}_n = \kappa_n / \sigma^n$. Substituting $\kappa_n = \hat{\kappa}_n \sigma^n$ and using the definitions $w_a = (\sigma_c/\sigma_a)^2$ and $w_b = (\sigma_c/\sigma_b)^2$:

$$\hat{\kappa}_{n,c}\sigma_c^n \approx \left( \frac{\sigma_c^2}{\sigma_a^2} \right)^n \hat{\kappa}_{n,a}\sigma_a^n + \left( \frac{\sigma_c^2}{\sigma_b^2} \right)^n \hat{\kappa}_{n,b}\sigma_b^n. \tag{50}$$

Dividing by $\sigma_c^n$ gives the final contraction relationship:

$$\hat{\kappa}_{n,c} = \left(\frac{\sigma_c}{\sigma_a}\right)^n \hat{\kappa}_{n,a} + \left(\frac{\sigma_c}{\sigma_b}\right)^n \hat{\kappa}_{n,b} + O(\epsilon^2). \tag{51}$$

$$= \left(\frac{\sigma_b^2}{\sigma_a^2 + \sigma_b^2}\right)^{n/2} \hat{\kappa}_{n,a} + \left(\frac{\sigma_a^2}{\sigma_a^2 + \sigma_b^2}\right)^{n/2} \hat{\kappa}_{n,b} + \mathcal{O}(\epsilon^2). \tag{52}$$

Since $\sigma_c < \sigma_a$ and $\sigma_c < \sigma_b$ (precision addition), the weights $(\sigma_c/\sigma)^n$ are strictly less than 1. Moreover, we neglect terms of $O(\epsilon^2)$ under the assumption that the system operates within the convergence regime ($R > 6$) identified in Lemma 4.2, where $\epsilon$ is actively suppressed by the path topology, thereby proving the reduction of non-Gaussianity.

# F. Proof of Theorem 4.4

In this Section, we provide the complete derivation for Theorem 4.4. While Theorem 4.1 established Gaussian convergence for simple paths, general tree topologies introduce branching paths where messages are multiplied at variable nodes. We prove that this structure does not prevent Gaussian emergence by decomposing the tree into paths originating from high-confidence priors (as identified in Lemma 4.2). By combining the convolutional decay along these paths with the stability of message multiplication in the near-Gaussian regime (Lemma 4.3), we demonstrate that the belief at any target node converges to a Gaussian distribution as the topological distance from the nearest strong data constraint increases.

In a tree topology, the belief at a target variable $X_t$ is the product of all incoming messages from its neighbors $\mathcal{N}(X_t)$:

$$b(X_t) \propto \prod_{k \in \mathcal{N}(X_t)} m_{k \to t}(X_t) \tag{53}$$

We analyse the distribution of these incoming messages based on their origin.

Let $\mathcal{P}$ be the set of all prior factors in the graph. We define the "closest prior" for the target $X_t$ as the prior $p^* \in \mathcal{P}$ that minimizes the topological distance $d(X_t, p)$. As established in Lemma 4.2, highly confident, non-Gaussian priors dominate the local belief product, violating the small-$\varepsilon$ assumption of Lemma 4.3. These priors therefore act as boundary conditions that "reset" the non-Gaussianity of the belief.

Consider any path segment of length $d$ originating from such a reset point (a strong prior) and terminating at $X_t$. Along this path, the message update consists exclusively of convolution with pairwise potentials (interspersed with multiplication by weak or uniform messages from side branches). By Theorem 4.1, the sequence of convolutions drives the message distribution toward a Gaussian, causing the standardized cumulants $\hat{\kappa}_n$ to decay as $d$ increases.

At any variable node $X_v$ along the path where branches merge, the outgoing message is the product of incoming messages. Provided the distance $d$ from the reset point is sufficient such that the cumulants have decayed to being near-Gaussian ($\varepsilon^2 \ll \varepsilon$), Lemma 4.3 guarantees that the multiplication of these messages introduces a perturbation of only $\mathcal{O}(\varepsilon^2)$. Since the convolutional decay of $\varepsilon$ (first order) strictly dominates the multiplication error $\mathcal{O}(\varepsilon^2)$ (second order), the convergence to a Gaussian distribution is stable.

**Conclusion**

Consequently, as the topological distance $d$ from the nearest strong prior increases, the dominant component of the belief becomes Gaussian, with non-Gaussian distortions decaying asymptotically.

# G. Proof of Theorem 4.5

We reproduce the classical computation-tree equivalence for Synchronous BP; our proof follows Weiss (2000).

Let $G = (V, F)$ be the loopy pairwise factor graph, where $V$ denotes the set of variable nodes and $F$ denotes the set of factor nodes.

Index synchronous BP iterations by $t = 0, 1, 2, ...$, and write messages between nodes $\alpha, \beta \in V \cup F$ as $m_{\alpha \to \beta}^{(t)}$. In synchronous BP, each update uses only messages from the previous iteration, so $m_{\alpha \to \beta}^{(t)}$ depends solely on collections of non-reversing paths of length $t$ ending at the edge $(\alpha, \beta)$.

Construct the computation tree $T_x^{(n)}$ by unwrapping all non-reversing walks in $G$ starting at $x$ up to length $n$, and assign to its leaves the same boundary messages as the initialization in $G$. By construction, every radius-$t$ non-reversing neighbourhood around any directed edge in $G$ has a unique isomorphic copy around the corresponding edge(s) in $T_x^{(n)}$.

We prove by induction on $t$ that for every directed edge $e$ in $G$ whose copy appears in $T_x^{(n)}$ at depth at most $t$, the tree message $\tilde{m}^{(t)}$ on the corresponding edge equals $m^{(t)}$ on $G$. The base $t = 0$ holds by the shared initialization on $G$ and on the tree leaves.

For the inductive step, the BP update at iteration $t$ is a deterministic function of the collection of incoming messages from iteration $t - 1$; by the induction hypothesis and the neighbourhood isomorphism, those incoming messages match on $G$ and on $T_x^{(n)}$, hence the updated message matches as well.

Beliefs are products of the most recent incoming messages at a node. If $u$ sits at graph distance $d \leq n$ from $x$ in $G$, then the messages that determine $b_u^{(n-d)}$ come from directed paths of length at most $n - d$; the same set of paths appears around the copy $\hat{u}$ at depth $d$ in $T_x^{(n)}$. Applying the induction with $t = n - d$ gives $b_u^{(n-d)} = \tilde{b}_{\hat{u}}^{(n-d)}$ as claimed.

This is the standard computation-tree ("unwrapped network") argument; interested readers are directed to Weiss (2000) for the original development and further consequences.

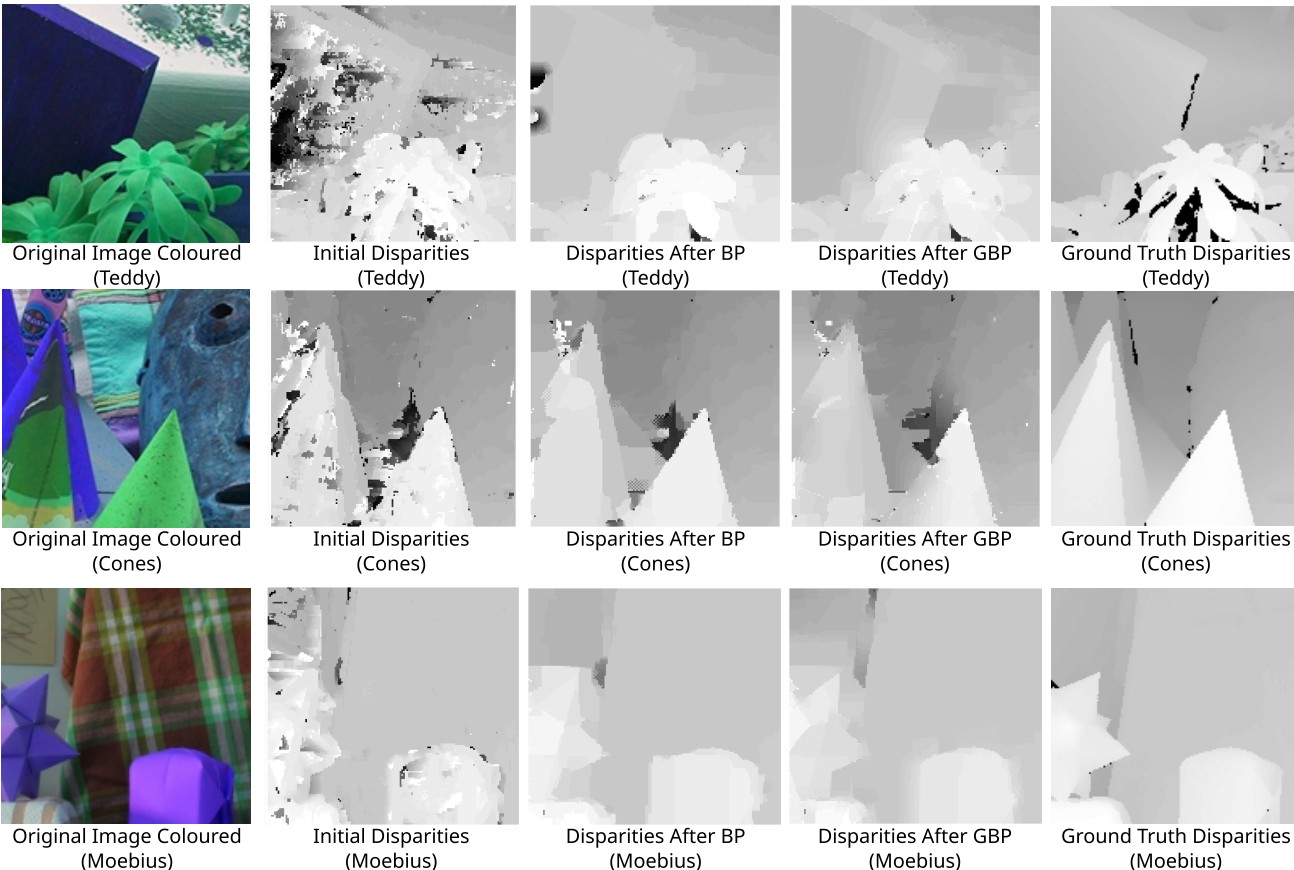

*Figure 7.* **BP & GBP are effective optimisers for stereo depth estimation across a range of scenes.**

## H. Additional Experimental Results

This Section details the additional experimental results achieved across different real-world images, comparing BP and GBP. All images are from the Middlebury stereo datasets - "Teddy" and "Cones" are from Scharstein & Szeliski (2003), while "Moebius" is from Scharstein & Pal (2007).

**Setup.** We use the same stereo depth estimation protocol, hyperparameters, and evaluation metrics as in the main paper, with additional scenes from the Middlebury family (e.g. Moebius). Implementation details are provided in Appendix I.

**GBP/BP are strong optimisers for stereo depth.** Across all additional scenes, both GBP and BP reliably decrease the MSE when compared against the ground truth. Fig. 7 demonstrates visually that the results achieved are reasonable approximations of the ground truth. Furthermore, the optimisation traces in Fig. 8 show monotonic improvement from the initialisation to a stable converged value.

**Beliefs become approximately Gaussian after BP.** Figure 9 reports the KL divergence between the distribution of the variable beliefs and a fit Gaussian. We see this metric decreases for pixels in image regions with weak (high-variance) priors, and converges to a small value. This is expected from the main text, indicating that the Gaussian approximation is not only convenient but predictive of the algorithm's late-stage dynamics.

**Final MSE converges to a similar point for BP and GBP, even for non-Gaussian problems.** Despite the highly non-Gaussia problem structure and the implementation across several scenes, the converged MSE for BP and GBP are nearly indistinguishable (see Fig. 8), and moreover, that GBP can often converge to a more optimal MSE value.

Taken together, these results reinforce that (i) GBP is a robust surrogate for BP in a surprisingly large number of cases, and

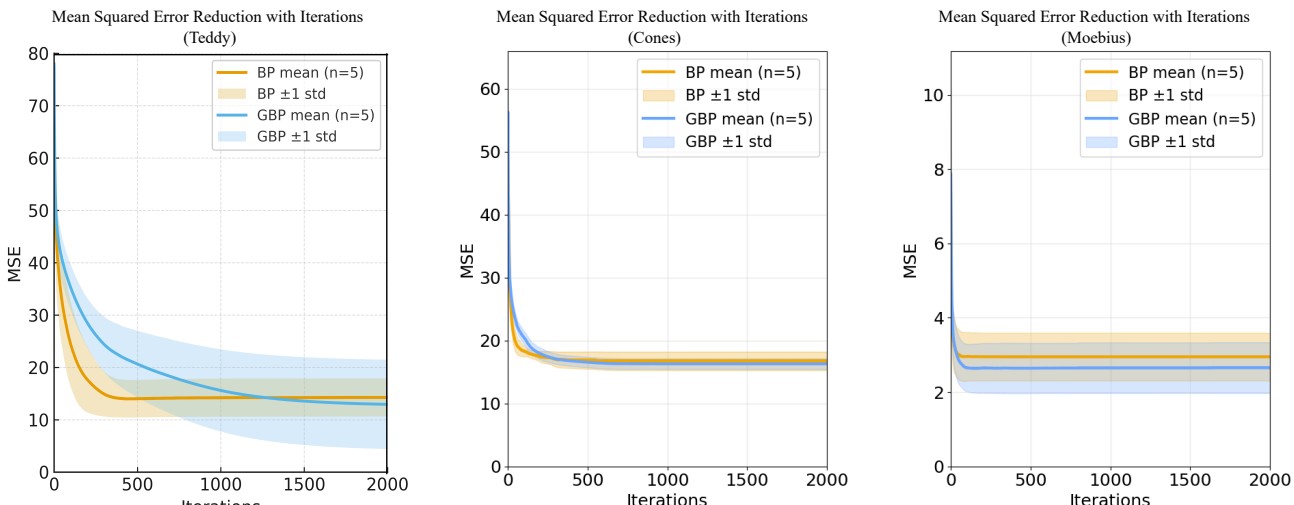

*Figure 8.* **Across a range of scenes, both BP and GBP converge to similar results for complex non-Gaussian problems.**

(ii) the variable beliefs are dominated by Gaussian interactions for both BP and GBP.

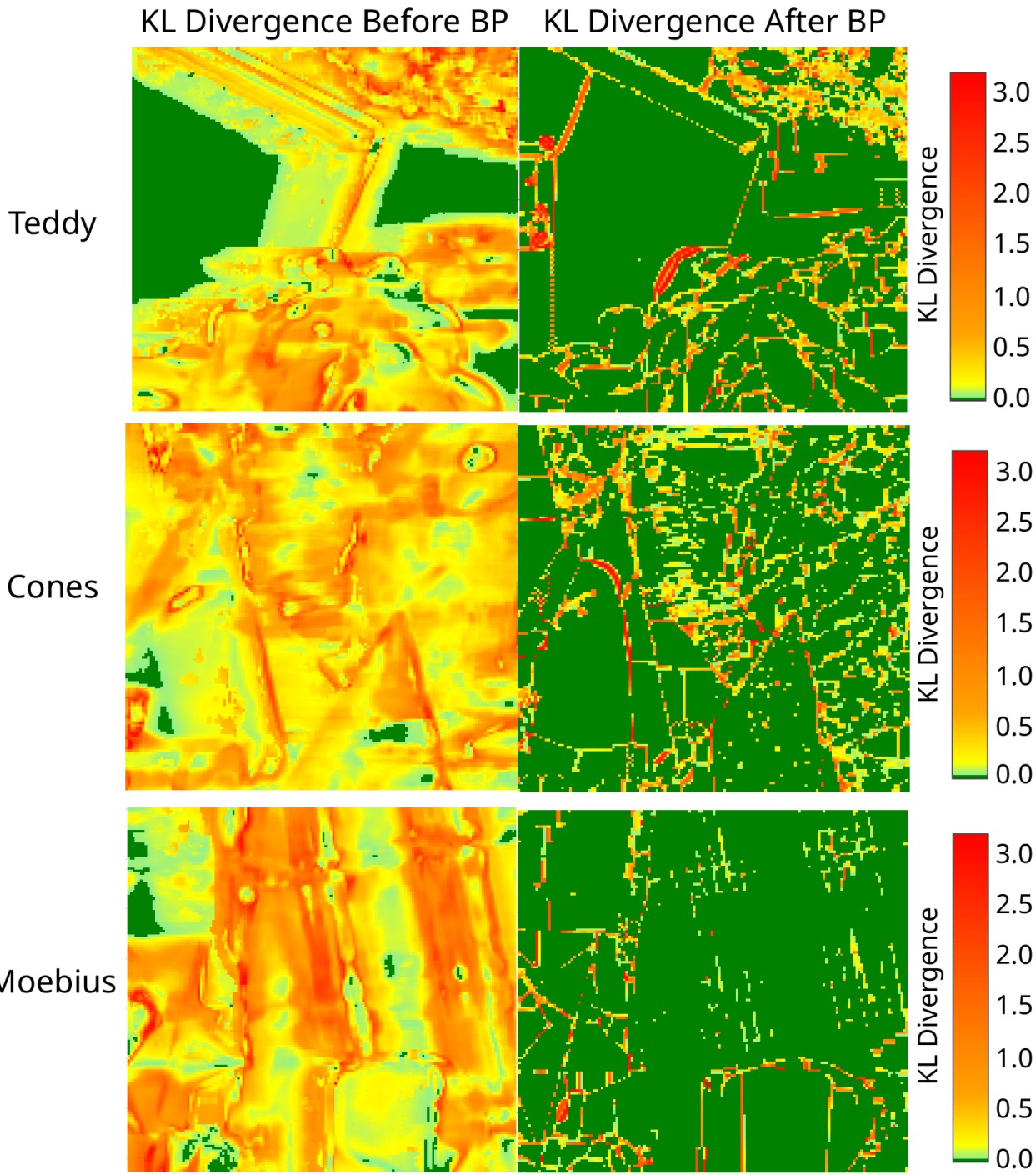

*Figure 9.* **Under Assumptions 1.1–1.4, BP causes variable beliefs to converge to a Gaussian distribution for a range of scenes.** This is measured by the reduction in KL Divergence between pixel beliefs and a fit Gaussian after BP has converged to a final state.

# I. Implementation Details

HARDWARE

All experiments are run locally with the hardware specifications below.
CPU = Intel Core Ultra 5 245KF 14 Core LGA1851 Processor
RAM = Corsair Vengeance DDR5 32GB (2x16GB) 5600Mhz
Memory = 2x WD Black SN850X 2TB NVMe PCIe 4.0 Solid State Drive

TUNEABLE PARAMETERS

There were several hyperparameters that required tuning throughout the experiments included in this work. Their values are detailed below, for reproducibility.

*Parameters used for Convergence Rate Experiment (Fig 4a*
Path number of variables = 13
Tree number of variables = 127
Grid number of variables = 16

Path number of prior factors = 2
Tree number of prior factors = 64
Grid number of prior factors = 1

Variable Belief Discretisation = 1024
Minimum measurement = -32
Maximum measurement = 31
Pairwise kernel distribution = random noise, 12 bins wide
Prior distributions = random noise, 16 bins wide
Random seed values = varying from 42, 142

*Parameters used for Node Degree Experiment (Fig 4b*
Number of variables = varying from 3-11
BP iterations = 30
Random seed values = varying from 42 to 72
Variable belief discretisation = 1024 bins
Minimum measurement = -32
Maximum measurement = 31
Prior distributions = random noise, 30 bins wide
Pairwise kernel distribution = random noise, 12 bins wide

*Parameters used for Prior Strength Experiment (Fig 4c*
Number of variables = 20
BP/GBP iterations = 20
Variable belief discretisation = 128 bins
Minimum measurement = 0
Maximum measurement = 63
Pairwise kernel distribution = random noise, 8-bins wide
Prior distribution = bounded uniform distribution, varying from 1-bin to 128-bins wide
Random seed values = varying from 42-142

*Parameters used for Stereo Depth Experiments (Figs 4d, 6, 7, 8 & 9)*

Image size = 150 * 200 pixels
Patch size = 5 pixels * 5 pixels
Lambda = 0.002
Edge mask disparity threshold = 3
Number of variables = 30,000
BP/GBP iterations = 2,000
Random seed values = 42-47

