# OpenReview forum: "Belief Propagation Converges to Gaussian Distributions in Sparsely-Connected Factor Graphs"
_ICML.cc/2026/Conference — ICML 2026 regular_

### Official Review · Reviewer_jwKZ · 2026-03-01

**Soundness:** 3
**Presentation:** 2
**Significance:** 2
**Originality:** 2
**Overall Recommendation:** 4
**Confidence:** 1

**Summary:**

This paper investigates the theoretical properties of Belief Propagation (BP) in sparsely-connected, non-Gaussian factor graphs, which are frequently encountered in spatial AI and robotics. The authors aim to provide theoretical guarantees for when Gaussian Belief Propagation (GBP) is a valid approximation in these settings. The core theoretical contribution relies on the Central Limit Theorem (CLT) to demonstrate that repeated convolutions along graph paths drive variable beliefs toward Gaussian distributions. The authors provide bounds for when highly confident prior factors prevent this convergence ($R \le 6$) and prove that message multiplication at variable nodes introduces negligible non-Gaussian perturbations under certain conditions. Although the paper provides an elegant theoretical explanation for the emergence of Gaussian distributions in sparsely-connected non-Gaussian graphs, its theoretical bounds (e.g., $R \le 6$) are intractable to compute a priori in complex, real-world SLAM systems. Furthermore, as this framework fails to translate into actionable guidance for algorithmic performance improvement, the work currently remains confined to a post-hoc explanation, limiting its practical engineering contribution to the field of Spatial AI.

**Compliance With Llm Reviewing Policy:**

Affirmed.

**Final Justification:**

I am not sure

**Key Questions For Authors:**

Regarding Lie Groups and High Uncertainty: Appendix A states that the Lie group extension holds for "concentrated distributions" where group operations map to vector addition in the tangent space. How do you reconcile this "small noise" requirement with the paper's primary claim of handling highly non-Gaussian, large-uncertainty problems? At what noise scale does the tangent space approximation break down and invalidate Theorem 4.1?

Regarding the CLT Independence Requirement: In unwrapped loopy graphs (Theorem 4.5), cyclic messages contain redundant information. How does your derivation account for the violation of the random variable independence assumption (required by the CLT)  when noise sources are repeatedly sampled from the same physical factors in tight loops?

Regarding Lemma 4.2 ($R \le 6$): The derivation in Appendix D assumes a worst-case scenario where non-Gaussian cumulants align constructively. In typical spatial AI applications, how often does destructive interference naturally allow for Gaussian convergence even when $R \le 6$?

Beyond providing a theoretical peace-of-mind, how can an engineer actually use your findings to design a better SLAM back-end? Does this theory prescribe a specific metric that can be computed in real-time to dynamically switch between GBP and non-parametric BP? I strongly urge the authors to explicitly discuss the algorithmic utility of their theory (e.g., guiding hybrid inference or graph sparsification strategies) to justify its impact on the Machine Learning and Robotics communities.

**Limitations:**

While the paper makes a commendable theoretical contribution by proving how Gaussian distributions emerge in sparsely-connected, non-Gaussian factor graphs, its practical contribution to the field of Spatial AI (such as SLAM) remains significantly limited.

The authors successfully provide a post-hoc theoretical explanation for certain phenomena observed in existing systems (e.g., explaining why GBP remains stable in textureless regions where priors are weak, and why it gets "anchored" at high-contrast edges). However, there is a massive gap between explaining a phenomenon and utilizing that theory to actively improve or optimize a system.

Specifically, the current theoretical framework and empirical results struggle to support proactive algorithmic improvements for SLAM systems, due to the following reasons:

Infeasibility of Preemptive Detection: The theoretical boundary for optimization failure (the "Exclusion Zone", $R \le 6$) requires knowing the exact variance of the non-Gaussian prior ($\sigma_{prior}^2$). In a real-world SLAM front-end encountering severe perceptual aliasing, accurately pre-computing the variance of such a highly multi-modal distribution is computationally prohibitive without expensive sampling—which defeats the entire purpose of using a lightweight GBP solver.

Topological Complexity vs. Theoretical Ideal: The critical threshold $R \le 6$ is analytically derived for a 1D chain graph. In a real 3D SLAM pose-graph with complex loop closures and varying node degrees, this threshold would likely fluctuate unpredictably, making it nearly impossible to set a static threshold to detect or prevent system collapse.

Conclusion on Impact: Because the proposed theory does not currently prescribe an actionable algorithmic strategy (e.g., it does not guide dynamic graph sparsification, nor does it propose a real-time metric for switching to hybrid inference to achieve lower latency or higher accuracy), it serves more as a theoretical "peace-of-mind" rather than an engineering tool. This inability to translate theoretical bounds into tangible performance gains or robustness improvements in a real SLAM pipeline severely limits the paper's overall contribution to the applied machine learning and robotics communities.

**Strengths And Weaknesses:**

**Strengths:**

Novel Perspective on an established Algorithm: The paper addresses a significant theoretical gap. While GBP is widely used in spatial AI (e.g., robust estimation, SLAM), its application to highly non-Gaussian problems has largely relied on empirical success rather than theoretical justification.

Insightful Distinction from Dense Graphs: The authors provide a compelling theoretical distinction between their "sparse" regime and the traditional "large system limit" found in coding theory. The observation that Gaussianity emerges from topological depth (convolution) rather than nodal averaging (which is shown to actually increase non-Gaussianity) is a strong theoretical insight.

Comprehensive Theoretical Framework: The progression of proofs from chain graphs (Theorem 4.1) to trees (Theorem 4.4) and finally to loopy graphs (Theorem 4.5) is logical and mathematically well-structured, supported by detailed derivations using characteristic functions and cumulant expansions.

**Weakness:**

Assumption 1.4 Limits Applicability to Highly Non-Linear Systems: Assumption 1.4 requires factor potentials to be shift-invariant, relying only on the difference $x_i - x_j$. In Appendix A, the authors argue this extends to Lie groups ($SO(3)$, $SE(3)$) via the tangent space. However, this tangent-space mapping explicitly assumes that distributions are concentrated (small noise relative to curvature). This fundamentally contradicts the paper's core premise of analyzing "highly non-Gaussian" problems with "highly uncertain data terms". If the noise is large enough to be non-Gaussian, the linear approximation on the manifold fails.

Independence Assumption in Loopy Graphs: Theorem 4.5 relies on Weiss's computation tree equivalence to unwrap loopy graphs. While mathematically valid for message values, the unwrapped tree duplicates variables extensively. The Lindeberg-Feller CLT fundamentally requires independent random variables. In tight loopy graphs (like the grid used in stereo depth), paths quickly become highly correlated. The paper lacks a rigorous discussion on how this correlation affects the independence assumption required for the convolutional CLT mechanism.

Lack of Diverse Real-World Baselines: For a paper claiming relevance to spatial AI , evaluating solely on 2D image stereo depth estimation is insufficient. The paper conspicuously lacks experiments on 3D pose-graph optimization or SLAM datasets, which would be necessary to substantiate the claims made in Appendix A.

It remains challenging to extract clear practical implications for downstream tasks from the current version.

---

> ### Author Rebuttal · Authors · 2026-03-30
>
> We thank the reviewer for their rigorous feedback - they have identified important boundaries to our results, and the new experiments helped us clarify these points more precisely:
>
> **1. Topological Complexity vs Theoretical Ideal for $R\le6$**
>
> We agree that the $R\le6$ threshold is derived for an idealised chain and therefore provides a conservative boundary for the simplest topology.
>
> To assess how this behaves in more complex graphs, we ran an additional experiment on fixed-depth regular trees and regular loopy graphs, defining an empirical $R^*$ threshold as the smallest value of $R$ for which the measured belief meets the same KL criterion used elsewhere ($D_{KL}<0.02$). We find that $R^\*$ varies only weakly with node degree in both topology families, with a modest uplift for loopy graphs. This suggests that once messages are near-Gaussian after a few hops (Fig 4a), the $R$ threshold depends only weakly on the precise topology.
>
> We will update the camera-ready version accordingly: $R\le6$ should be interpreted as a conservative path-based guideline, while $R^*$ provides the more practically relevant empirical threshold, which in our experiments remains close to the path-based value.
>
> **2. Infeasibility of Pre-emptive Detection for $R$**
>
> We agree that in a fully black-box SLAM front-end, computing the variance of arbitrary, highly multi-modal factors may be impractical. Our intended use case is instead for variances that are known _a priori_. For instance, in distributed sensor networks, hand-specified smoothness models, or SLAM systems with robust estimators (like Huber kernels), the variance is a hyperparameter. In these settings, $R$ is directly available and can guide when a Gaussian approximation is reliable.
>
> More generally, the theory gives a useful screening heuristic for when GBP is expected to be reliable in factors with known scale parameters, even if it does not yet yield a universal online switching criterion. We will add a 'Limitations' paragraph to the Conclusion, discussing the infeasibility of dynamically using $R$ for unknown factor distributions.
>
> **3. Lie Groups and High Uncertainty**
>
> We agree there is a tension between the asymptotic limit of the Euclidean CLT (where variance grows indefinitely) and the tangent-space approximation for Lie groups (which requires distributions to remain localised). Theorems 4.1-4.5 are proven only on $\mathbb{R}^d$, and our empirical validation is restricted accordingly.
>
> We do think the mechanism remains relevant in the practically important intermediate regime of robust estimation on manifolds, where the non-Gaussian factors are localised enough ("concentrated distributions") to remain well represented in tangent space. However, this is not covered by our current guarantees.
>
> In the final version, we will revise Section 1.1 and Appendix A to clarify and explicitly state that the extension to Lie groups is limited to these "concentrated distributions", and acknowledge that global uncertainties on $SO(3)$ fall outside our guarantees but that they pose a valuable direction for future work.
>
> **4. Lack of Diverse Real-World Baselines**
>
> Since the current theorems are proven only in Euclidean space, we chose a real task in $\mathbb{R}^d$ to test the specific mechanism claimed by the paper. Extending the evaluation to 3D pose-graph optimisation on Lie groups would require addressing the tangent-space limitations above, and we agree that this is an important direction for future work. We will make this boundary clearer in the conclusion.
>
> **5. Lemma 4.2 and Destructive Interference**
>
> The reviewer is correct that Lemma 4.2 is derived under a worst-case constructive interference assumption for the leading non-Gaussian cumulants. Our new empirical result, where $R^*$ is slightly below 6, is therefore consistent with Appendix D which suggests destructive interference can allow Gaussian convergence at ratios lower than the idealised bound.
>
> We will revise Appendix D and the discussion around Lemma 4.2 to clarify that $R=6$ should be read as a conservative stability boundary for the idealised path, while $R^*$ provides the more practically relevant empirical threshold in graphs with priors at every variable.
>
> **6. The CLT Independence Requirement**
>
> We agree that in loopy graphs the original latent variables are not independent, and repeated traversal of the same factors induces correlations. Our CLT argument is therefore not a statement about the true variables themselves, it is a statement about the BP message-update operator. In synchronous loopy BP, the update can be analysed via its computation-tree, where repeated uses of a factor appear as distinct branches in the operator sequence. Loopy BP therefore treats these repeated factor potentials as independent, and this is the level at which the Lindeberg-Feller CLT is invoked. It also explains why Loopy BP is only approximate. We will revise Section 4.4 and Appendix G to make this distinction explicit.

---

> > ### Author Rebuttal · Reviewer_jwKZ · 2026-04-01
> >
> > "Lack of Diverse Real-World Baselines" this limitation should be worked out in the future. I will keep my score.

---

### Official Review · Reviewer_SS73 · 2026-03-10

**Soundness:** 3
**Presentation:** 3
**Significance:** 3
**Originality:** 3
**Overall Recommendation:** 4
**Confidence:** 3

**Summary:**

This article provides a theoretical justification for the successful application of Gaussian Belief Propagation to non-Gaussian graphical models. The results apply to sparsely-connected graphs satisfying a set of stated conditions. The authors first show for a factor graph being a path with a single prior factor that the beliefs at the variable nodes approach a Gaussian the further along the path the variable node is located. The previous statement is extended to factor graphs with multiple priors by introducing a condition wrt. the prior and pairwise distributions. Next, the statements are further generalized to tree graphs by identifying conditions under which the convergence to Gaussians is preserved. Finally, it is shown that the statements are also applicable to loopy factor graphs. Results are validated using two types of experiments: Firstly, the convergence to Gaussians is validated on synthetic instances. Secondly, the practical applicability of Gaussian Belief Propagation is shown by a comparison with Belief Propagation on a stereo depth estimation task.

**Compliance With Llm Reviewing Policy:**

Affirmed.

**Final Justification:**

I suggest to accept this article.

The only reason I select "weak accept" instead of "accept" is that impact means a lot these days in ML, and I am not fully convinced this article will fall into the category of moderate to high impact.

**Key Questions For Authors:**

- Consider using the term "path" instead of "chain", because the term "chain graph" has also been used with a different meaning in the context of graphical models.
- Some references need to be fixed (check for missing details, missing DOIs/URLs, capitalization, consistent use of abbreviations)

**Limitations:**

yes

**Strengths And Weaknesses:**

Strengths:
- The properties established appear to be technically correct. Although I have not had the time to check the proofs in detail, the main arguments appear to be correct (I have not found a single mistake, not even a minor issue). In addition, the proofs are well-structured, progressing from simple cases to more general settings while carefully introducing the conditions required for the results to hold.
- The article makes a relevant contribution to the theoretical foundations of graphical model inference.
- All assumptions are stated clearly, and the scope of all results is discussed objectively.
- The paper is well-written; intuitions and figures provided are helpful for understanding the individual steps as well as the overall structure.

Weaknesses:
- I find the theoretical contribution too complex for me to verify in detail in the reviewing process of a conference. Perhaps, the contents are better suited for a submission straight to a journal.

---

> ### Author Rebuttal · Authors · 2026-03-30
>
> We would like to thank the reviewer for their thoughtful feedback and time spent reviewing our proofs. In response to the specific concerns identified:
>
> > I find the theoretical contribution too complex for me to verify in detail in the reviewing process of a conference. Perhaps, the contents are better suited for a submission straight to a journal.
>
> We have targeted ICML because we believe the core results provide a principled explanation for the empirical success of GBP in practical ML and vision applications, and would be of interest to the ICML audience. However, we understand the reviewer's perspective regarding the theoretical density of the proofs.
>
> In the revision, we will clearly separate the intuitive algorithmic mechanism in the main paper from the formal proofs in the Appendix to make our results more accessible to the broader ICML audience.
>
> > Consider using the term “path” instead of “chain”, because the term “chain graph” has also been used with a different meaning in the context of graphical models.
>
> We agree that "path" avoids confusion with existing graphical model literature, and gives a more intuitive understanding of the mechanism being described. In the revised version, we will replace mentions of "chain" with "path" where appropriate.
>
> > Some references need to be fixed (check for missing details, missing DOIs/URLs, capitalisation, consistent use of abbreviations)
>
> We appreciate the reviewer catching this, and for the final version will ensure that all references are complete, properly capitalised and use consistent abreviations.

---

> > ### Author Rebuttal · Reviewer_SS73 · 2026-04-01
> >
> > I agree with the authors that ICML is a suitable outlet for this article.

---

### Official Review · Reviewer_E3bG · 2026-03-13

**Soundness:** 3
**Presentation:** 3
**Significance:** 4
**Originality:** 3
**Overall Recommendation:** 5
**Confidence:** 3

**Summary:**

This paper addresses a theoretically important problem in understanding belief propagation, namely, the approximate Gaussian nature of variable belief distributions in a non-Gaussian, sparsely connected graph. The Gaussian distributional aspect is important, as the authors explain briefly in the Introduction, for justifying the use of belief propagation in non-Gaussian problems; for example, the validity of Gaussian approximation to beliefs in densely connected factor graphs is key to understanding convergence of the so-called "approximate message passing" algorithms. The key assumptions (Assumptions 1.3 and 4) are that factors are either unary or binary, and that binary factors are "stationary" (i.e., only dependent on the pairwise displacement between the variables). Under these restrictions, main findings are that the distributions of variable beliefs converge to a Gaussian distribution in a loopy, pairwise connected factor graph (Theorem 4.5). The authors validate their theoretical findings through an empirical study of the distribution of variable beliefs in a stereo depth estimation problem.

**Compliance With Llm Reviewing Policy:**

Affirmed.

**Final Justification:**

The minor issues of presentation I have raised have been addressed, and I have no further follow-up questions on the substance of the work. Despite the technical nature of the work, I believe it retains enough significance to warrant a publication in ICML.

**Key Questions For Authors:**

To what extent do you think the assumption of unary/binary factors can be relaxed, based on your current technical tools?

**Limitations:**

Yes.

**Strengths And Weaknesses:**

The authors make a convincing case that the addressed problem, while technical, is an important one for justifying the use of belief propagation in spatial AI applications. Empirical studies are also insightful, in that it raises an example in which a Gaussian distribution approximation is not valid. The weakness, of course, is the inapplicability of the theoretical results to a wider range of factor graphs, but this cannot be faulted on the authors. In terms of the exposition, I think the authors can provide a lengthier description of, and a clearer contrast with, the case of dense graphs in the Introduction, laying out the different mechanisms by which the central limit takes place in the two regimes (which is already somewhat included in Section 5).

---

> ### Author Rebuttal · Authors · 2026-03-30
>
> We would like to thank the reviewer for their thoughtful feedback. In response to the specific queries:
> > To what extent do you think the assumption of unary/binary factors can be relaxed, based on your current technical tools?
>
> Our current technical tools extend most directly to certain structured high-degree factors, rather than arbitrary ones.
>
> In particular, linear high-degree factors of the form $\phi(\mathbf{x})=g(\mathbf{x}\cdot\mathbf{v})$ can be decomposed into a sequence of unary/binary interactions using auxiliary variables (Potetz & Lee, 2008), reducing the update to recursive 1D convolutions. Our results are then expected to extend exactly to linear high-degree factors, though this requires empirical validation.
>
> For nonlinear high-degree factors, a first-order linearisation suggests the same mechanism may apply approximately, but this introduces operating-point-dependent error and is not covered by our present guarantees.
>
> We will therefore soften the discussion in the final version: the unary/binary assumption is exact and validated for the current theorems, while extension to high-degree factors presents a promising direction for future work.
>
> > I think the authors can provide a lengthier description of, and a clearer contrast with, the case of dense graphs in the Introduction, laying out the different mechanisms by which the central limit theorem takes place in the two regimes (which is already somewhat included in Section 5).
>
> We accept the reviewer's feedback, and for the revised version would include a paragraph offering further clarification into the Introduction.
>
> While the emergence of Gaussianity in BP has been well-studied in the large-system limit, such as CDMA and compressed sensing (Guo & Verdú, 2005; Donoho et al., 2009), it is driven by an entirely different mechanism in our setting.
>
> In the dense regimes, individual factor nodes are assumed to have infinite degree, representing linear mixtures of many variables. The CLT thus occurs locally at a single factor node, as marginalizing over an infinite sum of incoming independent variable messages instantly produces a Gaussian outgoing message.
>
> In contrast, our models operate on the sparse factor graphs common to Spatial AI, where both variables and factors possess strictly finite, low degrees, making a local, single-step CLT impossible. Instead, we demonstrate that Gaussianity emerges sequentially in sparse graphs, driven by the repeated convolution of messages across topological depth.
>
> > The inapplicability of the theoretical results to a wider range of factor graphs, but this cannot be faulted on the authors.
>
> We thank the reviewer for their fair assessment, and agree that the strength of the assumptions restricts the applicable range of factor graphs. For the camera-ready version, we will add a limitations paragraph to the Conclusion to discuss these boundaries and propose future work aimed at relaxing them.

---

> > ### Author Rebuttal · Reviewer_E3bG · 2026-04-02
> >
> > I thank the authors for their response. The minor issues of presentation I have raised have been addressed, and I have no further follow-up questions on the substance of the work. I am retaining my original score, which is to accept the paper.

---

### Official Review · Reviewer_Zpp7 · 2026-03-14

**Soundness:** 3
**Presentation:** 2
**Significance:** 2
**Originality:** 3
**Overall Recommendation:** 4
**Confidence:** 2

**Summary:**

This paper investigates Gaussian Belief Propagation in sparse, non-Gaussian factor graphs. The authors argue that, under a set of assumptions including finite moments, low-degree unary and pairwise factors, repeated message passing drives beliefs toward Gaussian distributions. The analysis is developed progressively for chains, trees, and loopy graphs, and is supported by synthetic experiments as well as a stereo depth estimation example. Overall, the paper aims to provide a principled explanation for the empirical success of GBP in sparse inference settings.

**Compliance With Llm Reviewing Policy:**

Affirmed.

**Final Justification:**

My concerns are solved. I will keep my current rating.

**Key Questions For Authors:**

1. The independence discussion in Section 4.1 argues via convolutional equivalence to sums of independent random variables. Can the authors clarify more carefully whether the theorem is a statement about BP as an operator sequence, rather than about the original latent variables being independent?

2. How sensitive is Lemma 4.2’s threshold R to heterogeneity in factors and graph structure? Some additional intuition or empirical calibration would help.

3. The real-data validation focuses mainly on Middlebury Cones, with mention of additional scenes in the appendix. Could the authors provide a broader summary of how consistently the Gaussian-emergence phenomenon appears across scenes and parameter settings?

**Limitations:**

The current limitations/impact discussion is not adequate. The paper should more explicitly discuss the limitations raised by assumptions.

**Strengths And Weaknesses:**

Strengths

1. The paper addresses a meaningful problem. Understanding when GBP is justified beyond the fully Gaussian case is important for applications in factor-graph inference.

2. The theoretical development is well organized. The progression from chains to trees to loopy graphs is a logically reasonable structure, and validates the method in increasingly general topologies.

3. The paper makes a commendable effort to connect theory with experiments. The author includes formal analysis, synthetic studies of convergence behavior, and a stereo depth estimation which helps ground the theory in a practical inference task.

4. The empirical section shows positive results and analysis of degree effects and prior strength. This makes the experimental section more informative than a purely benchmarking.

Weaknesses

1. The theoretical results depend on fairly restrictive assumptions, especially the shift-invariance of pairwise factors and the low-degree structure of the graph. While these assumptions may be reasonable for some spatial AI problems, they substantially narrow the practical scope of the claims.

2. Some parts of the analysis feel more heuristic than fully rigorous in presentation. In particular, the discussion connecting convolution to sums of independent random variables is intuitively useful, but the manuscript should distinguish more carefully between an analytical property of BP message updates and a probabilistic statement about the original latent variables in the graph.

3. The limitations discussion is too weak. The paper should be more explicit about the boundaries of the theory, especially in cases that violate the assumptions.

---

> ### Author Rebuttal · Authors · 2026-03-30
>
> We thank the reviewer for their thoughtful feedback.
>
> > How sensitive is Lemma 4.2’s threshold R to heterogeneity in factors and graph structure?
>
> On graph structure, we ran an additional experiment measuring an empirical R threshold, $R^\*$, for regular trees and loopy graphs, while varying node degree from 2-10. We define $R^\*$ as the smallest value of $R$ for which the measured belief meets the same KL criterion used elsewhere ($D_{KL}<0.02$ bits). In the settings tested, $R^*$ varies only weakly with node degree for each topology. A likely explanation is that messages become near-Gaussian in only a few hops (Fig 4a), after which multiplying messages preserves Gaussianity to first order (Lemma 4.3).
>
> On heterogeneity, we expect the threshold to be robust. Appendix D relies on the Lindeberg-Feller CLT, which accommodates non-identical distributions, and the threshold depends on a local variance ratio rather than homogeneity. This suggests heterogeneity should not significantly change $R^*$, though it requires empirical validation.
>
> In the revision we will include the topology experiment, alongside a second experiment varying $\sigma$ at factors in a chain to empirically validate $R$'s sensitivity to heterogeneity.
>
> > Can the authors clarify more carefully whether [Theorem 4.1] is a statement about BP as an operator sequence, rather than about the original latent variables being independent?
>
> We thank the reviewer for highlighting this distinction and will update Section 4.1 accordingly. Theorem 4.1 is a statement about the asymptotic behavior of the BP operator sequence rather than the independence of the latent variables, which are highly correlated.
>
> In chain/tree graphs, each sensor introduces noise that is independent of other sources. The mechanism driving Gaussian convergence is the accumulation of this noise, defined by the factor potentials. When BP constructs a message, it sequentially convolves the incoming message with these independent potentials. Because the algorithm is convolving independent noise kernels, the operations map to the sum of independent random variables, satisfying the Lindeberg-Feller CLT.
>
> As we will clarify in Theorem 4.5's revision, loopy graphs may mean noise sources are traversed repeatedly and become correlated. However, synchronous BP unwraps the graph into a computation tree, treating every reused factor as an independent random variable. This independence assumption then drives the message to be Gaussian via the CLT.
>
> > The paper should more explicitly discuss the limitations raised by assumptions.
>
> We agree, and will include a paragraph in the Conclusion clarifying the boundaries of our results. In particular, our current guarantees do not directly extend to:
> 1. non-euclidean state spaces (e.g. Lie Groups), except in the concentrated regime where tangent-space approximations remain valid
> 2. factors of degree greater than unary/binary, beyond linear cases of the form $\phi(\mathbf{x})=g(\mathbf{x}\cdot\mathbf{v})$ (which reduce to recursive 1D convolutions via auxiliary variables)
> 3. asynchronous or adaptive message schedules
> 4. loopy graphs that do not converge to steady state beliefs under synchronous BP
> 5. graphs where factor distributions are not known, so $R$ cannot be evaluated directly
>
> > Could the authors provide a summary of how consistently the Gaussian-emergence phenomenon appears across scenes and parameter settings?
>
> We see in Appendix H that Gaussian emergence is consistent across the 3 scenes tested. Gaussian heat maps show convergence to Gaussian distributions in textureless regions, while preserving the non-Gaussian distributions of highly textured regions (Fig 5, Appendix Fig 8). Moreover, the total Mean Squared Error converges to remarkably close values between BP and GBP across all scenes. These results imply that variable beliefs are dominated by Gaussian interactions in BP for a range of practical Spatial AI tasks.
>
> On parameter sensitivity, we found Gaussian emergence to be robust to variations in graph size and factor shape, provided a few key conditions are met. First, beliefs' discretisation resolution must be sufficient to faithfully represent a Gaussian (e.g. $\ge$ 20 bins). Second, the graph must have enough topological depth for the required message passing to enable the CLT (e.g. ~2-3 hops, Fig 4a). Third, although the asymptotic limit of the CLT allows variance to accumulate indefinitely, real discrete distributions have finite support; therefore, pairwise potentials must be kept narrow to avoid distribution truncation.
>
> Beyond these practical requirements, the main sensitivity is the relative strength of the local prior and pairwise factors, as captured by $R$. Our new topology experiment suggests that once messages have entered the near-Gaussian regime, the empirical threshold depends only weakly on topology.
>
> In the revision, we will add a paragraph to Section 5 summarising scene consistency and parameter sensitivity.

---

> > ### Author Rebuttal · Reviewer_Zpp7 · 2026-04-03
> >
> > My concerns are solved. I will keep my current rating.

---

### Decision · Program_Chairs · 2026-04-30

**Decision:**

Accept (regular)

**Comment:**

This paper addresses an interesting and relevant question: why Gaussian belief propagation can still work well in non-Gaussian sparse factor graphs. The reviews were uniformly positive, and the rebuttal clarified several aspects of the intended mechanism, the assumptions, and the empirical scope. The paper proposes a plausible Gaussianization mechanism along paths in sparse graphs and supports it with suggestive experiments.

My main reservation is that the theory seems somewhat stronger than what is fully established, especially for the tree and loopy extensions. In my reading, those parts are more suggestive than fully conclusive, and the related-work discussion could also be stronger, particularly in relation to the AMP/state-evolution literature. Overall, I am comfortable with this paper as a weak accept.